**Modeling demographic-driven vegetation dynamics and**
**ecosystem biogeochemical cycling in NASA GISS's Earth system**
**model (ModelE-BiomeE v.1.0)**
Ensheng Weng[1,2], Igor Aleinov[1,2], Ram Singh[1,2], Michael J. Puma[1,2], Sonali S. McDermid[3],
Nancy Y. Kiang[2], Maxwell Kelley[2], Kevin Wilcox[4], Ray Dybzinski[5], Caroline E. Farrior[6],
Stephen W. Pacala[7], Benjamin I. Cook[2]
[1]Center for Climate Systems Research, Columbia University, New York, NY 10025, USA
[2]NASA Goddard Institute for Space Studies, 2880 Broadway, New York, NY 10025, USA
[3]Department of Environmental Studies, New York University, New York, NY 10003, USA
[4]Department of Ecosystem Science and Management, University of Wyoming, Laramie, WY
82071, USA
[5]School of Environmental Sustainability, Loyola University Chicago, Chicago, IL 60660, USA
[6]Department of Integrative Biology, University of Texas at Austin, Austin, TX 78712, USA
[7]Department of Ecology & Evolutionary Biology, Princeton University, Princeton, NJ 08544,
USA
**Corresponding author:** Ensheng Weng (wengensheng@gmail.com; phone: 212-678-5585)
Accepted for publication in **Geoscientific Model Development**
**Abstract.** We developed a demographic vegetation model, BiomeE, to improve the modeling of
vegetation dynamics and ecosystem biogeochemical cycles in the NASA Goddard Institute of
Space Studies' ModelE Earth system model. This model includes the processes of plant growth,
mortality, reproduction, vegetation structural dynamics, and soil carbon and nitrogen storage and
transformations. The model combines the plant physiological processes of ModelE's original
vegetation model, Ent, with the plant demographic and ecosystem nitrogen processes that have
been represented in the Geophysical Fluid Dynamics Laboratory's LM3-PPA. We used 9 plant
functional types to represent global natural vegetation functional diversity, including trees,
shrubs, and grasses, and a new phenology model to simulate vegetation seasonal changes with
temperature and precipitation fluctuations. Competition for light and soil resources is individual-
based, which makes the modeling of transient compositional dynamics and vegetation succession
possible. Overall, the BiomeE model simulates, with fidelity comparable to other models, the
dynamics of vegetation and soil biogeochemistry, including leaf area index, vegetation structure
(e.g., height, tree density, size distribution, crown organization), and ecosystem carbon and
nitrogen storage and fluxes. This model allows ModelE to simulate transient and long-term
biogeophysical and biogeochemical feedbacks between the climate system and land ecosystems.
Further, BiomeE also allows for the eco-evolutionary modeling of community assemblage in
response to past and future climate changes with its individual-based competition and
demographic processes.

## 1 Introduction

Terrestrial ecosystems play a critical role in climate systems by regulating exchanges of energy, moisture, and carbon dioxide between the land surface and the atmosphere (Sellers, 1997; Pielke et al., 1998; Meir et al., 2006). In turn, climate change has significantly affected vegetation photosynthesis, water use efficiency, mortality, regeneration, and structure through gradual changes in temperature and atmospheric $CO_2$ concentration together with shifts in climate extremes (Keenan et al., 2013; Huang et al., 2015; Brando et al., 2019; McDowell et al., 2020). These responses have triggered structural and compositional shifts in global vegetation. For example, global forest mortality has increased in recent years  (Allen et al., 2010; Anderegg et al., 2012), tree sizes have decreased (Zhou et al., 2014; McDowell et al., 2020), and species composition has shifted to more opportunistic species (Clark et al., 2016; Brodribb et al., 2020). The shifts in vegetation function, composition, and structure can change the boundary conditions of the land surface and affect the climate system (Nobre et al., 1991; Avissar and Werth, 2005; Garcia et al., 2016; Green et al., 2017; Zeng et al., 2017). Realistic simulation of these processes is therefore critical for Earth system models (ESMs).

The vegetation dynamics in ESMs are usually simulated using dynamic global vegetation models (DGVMs) (Prentice et al., 2007), most of which are simplified in their representation of ecological processes. The core assumptions of many vegetation models are a big-leaf canopy, vegetation represented by only a few plant functional types (PFTs), single cohort-based vegetation dynamics ("single-cohort" assumption, where the vegetation community at a land unit are simulated as a collection of identical plants), lumped-pool-based biogeochemical cycles and first order decay of soil organic matter. The competition of plant individuals and vegetation types

is approximately simulated as a function of productivity or Lotka-Volterra equations to predict
fractional PFT coverage (e.g., SDVGM, HYBRID, TRIFFID) (Friend et al., 1997; Woodward et
al., 1998; Sitch et al., 2003). These simplifying assumptions make it possible to simulate the
complex interactions of biological and ecological processes at the global scale.
These models are generally successful in reproducing land surface carbon, energy, and
water fluxes after extensive tuning against data from sites, observational networks, and satellite
remote sensing. However, the uncertainty of model predictions is high, and predictions can
diverge substantially across different models (Friedlingstein et al., 2014; Arora et al., 2020).
Lack of functional diversity and community assembly processes is one of the key issues in the
vegetation modeling of ESMs, which makes the models unable to predict transient dynamics of
vegetation composition and structure. A more mechanistic design that uses the fundamental
principles of ecology to simulate the emergent properties of ecosystems for predicting ecosystem
dynamics may therefore be necessary (Scheiter et al., 2013; Weng et al., 2017).
To this end, extensive efforts have been made to improve the representation of transient
vegetation dynamics based on ecological theories and conceptual models. Two pivotal advances
have been made in ecological vegetation modeling: 1) Demographic processes and trait-based
representation of processes have been developed to improve the representation of functional
diversity and size (Pavlick et al., 2013; Fisher et al., 2015; Weng et al., 2015; Argles et al., 2020)
and 2) eco-evolutionary optimal and game theoretical approaches have been proposed to predict
the flexibility of parameters and processes (McNickle et al., 2016; Weng et al., 2017). These
concepts are mainly applied in modeling photosynthesis (Prentice et al., 2014; Wang et al.,
2017), allocation (Farrior et al., 2013; Dybzinski et al., 2015), and evolutionarily stable strategy
of plant traits (Falster et al., 2017; Weng et al., 2017). These ideas for incorporating ecological
and evolutionary principles into ESMs have been summarized in several recent review papers
(Franklin et al., 2020; Harrison et al., 2021; Kyker-Snowman et al., 2022).
There are still formidable challenges to integrating the sophisticated ecological modeling
approaches into land models, which explicitly simulate energy, water, and carbon fluxes at high
frequency time steps for interacting with the atmosphere and climate systems. Including highly
complex processes does not necessarily increase model predictive skills (Forster, 2017; Hourdin
et al., 2017; Famiglietti et al., 2021). On the contrary, it may greatly complicate model structure,
obscure model transparency, and increase model uncertainty; positive feedbacks in these
processes may result in large and unanticipated shifts of vegetation states. Any small differences
in model settings or parameters can result in distinct predictions, especially for vegetation
structure, which is supposed to be predicted by these types of models. Additionally, the long
history of model development and the requirements of backward compatibility (i.e., reversing the
model to its previous versions) mean developers often build their new functions on top of
previous modeling assumptions and coding structure (Fisher and Koven, 2020), adding up to
multiple adjustments of previous processes and making the model untraceable.
To explicitly simulate ecosystem transient dynamics in ESMs while preserving model
traceability, we need clear assumptions, detailed physical processes, and traceable model
structure. The details of vegetation processes, including plant physiological processes (e.g.,
photosynthesis and respiration), phenology, plant growth, reproduction, mortality, competition
for different resources, and community assembly, must be well-organized hierarchically and
computed efficiently (Fisher and Koven, 2020; Franklin et al., 2020). For the best chance of
accurate predictions outside of the model's testing data, model processes should be based on the
fundamental biological and ecological principles to predict ecosystem emergent properties,
instead of fitting the emergent patterns directly as many models do currently.

To achieve this, we need to properly represent plant functional diversity and tradeoffs of

plant traits, balance the complexity of the model structure and priority for the processes that are
required by ESMs (e.g., surface reflectance, drag coefficient, carbon and water cycles), and also
make model assumptions transparent and processes robust. These requirements make it difficult
to fully implement the modeling approaches that are well-developed in the ecological modeling
community (e.g., Falster et al., 2016; Berzaghi et al., 2019; Weiskopf et al., 2022). A
parsimonious approach of is necessary in the modeling of vegetation demographic processes and
population dynamics in ESMs.

In this paper, we describe a parsimonious terrestrial biosphere model that incorporates the

vegetation demographic and soil biogeochemical processes into the NASA Goddard Institute for
Space Studies (GISS) Earth system model, ModelE (Kelley et al., 2020). The major ecosystem
processes, such as plant growth, demography, community assembly, and ecosystem carbon and
nitrogen cycles are included in this model. These processes set up a framework for solving the
major challenges of modeling ecological mechanisms in ESMs and allow ModelE to simulate the
ecological dynamics of terrestrial ecosystems. In this paper, we describe this model in detail, and
evaluate its performance compared to both observations and other state-of-the-art DGVMs.

## 2 Model Description

### 2.1 GISS ModelE and BiomeE overview

ModelE has a land model for representing land surface hydrology (TerraE) (Rosenzweig and Abramopoulos, 1997; Schmidt et al., 2014) and a vegetation biophysics scheme (from the Ent Terrestrial Biosphere Model; Ent TBM) (Kim et al., 2015; Ito et al., 2020; Kelley et al., 2020), with fixed vegetation traits (e.g., leaf mass per area, C:N ratio), fixed biomass, canopy height, and plant density, and seasonal leaf area index prescribed from a satellite-derived data set (Ito et al., 2020). The Ent TBM calculates canopy radiative transfer (Friend & Kiang 2005), canopy albedo, canopy conductance, photosynthesis, autotrophic respiration, and phenological behaviors (Kim et al., 2015). The carbon allocation scheme of Kim et al. (2015) is used in ModelE with prescribed canopy structure and leaf area index (LAI), routing the carbon that would otherwise be allocated to plant tissues via growth instead directly as litter into soil carbon pools, thus conserving carbon for fully coupled carbon cycle simulations, but resulting possibly in imbalanced plant carbon reserve pools where the prescribed canopy structure is not in equilibrium with the simulated climate (Ito et al., 2020).

The Biome Ecological strategy simulator (BiomeE) is derived from the Geophysical Fluid Dynamics Laboratory's vegetation model, LM3-PPA (Weng et al., 2015, 2017, 2019). It simulates plant physiology, vegetation demography, adaptive dynamics (eco-evolutionary adaptation), and ecosystem carbon, nitrogen, and water cycles (Figure 1). In this model, the PFTs are defined by a set of combined plant traits with their values sampled from the observed ranges to represent a specific plant type. Individual plants are categorized into cohorts and arranged in different vertical canopy layers according to their height and crown area following

the rules of the Perfect Plasticity Approximation model (PPA, Strigul et al., 2008). Sunlight is
partitioned into canopy crown layers according to Beer's law (Beer, 1852; Swinehart, 1962). The
cohort is the basic unit to carry out physiological and demographic activities, e.g.,
photosynthesis, respiration, growth, reproduction, mortality, and competition with other
individuals.

The demographic processes generate and remove cohorts and change the size and density

of plant individuals in the cohorts. With explicit representation of cohort size and crown
organization, the model simulates competition for light and soil resources, community assembly
and vegetation structural dynamics.  These processes are hierarchically organized in this model
and run at various time steps: half-hourly or hourly for plant physiology and soil organic matter
decomposition, daily for growth and phenology, and yearly for demography.

For extending this model to the global scale, we designed a new set of PFTs to represent

the functional diversity of global vegetation and a new phenological scheme to deal with
temperature and water seasonality in coupling BiomeE into ModelE. Leaf photosynthesis
processes are taken from ModelE's existing vegetation model, Ent (Kim et al., 2015), and used
to calculate the carbon budget that drives vegetation dynamics. Plant growth, demographic
processes, and soil organic matter decomposition and nitrogen cycle processes are from BiomeE
(Figure 1). The land surface energy and water fluxes are calculated by TerraE with land surface
characteristics jointly defined by the vegetation model.
**2.2 Plant functional types**
In this model, we use a set of continuous plant traits to define plant functional types, so that the
model is able to predict vegetation emergent properties (such as dominant plant types, size
structure, compositional dynamics, etc.) in different climatic conditions based on the underlying
plant physiological properties and ecological principles through eco-evolutionary modeling in
the future. For example, life forms are defined by the continuums characterized by wood density
(woody vs. herbaceous), height growth coefficient (tree vs. shrub), and leaf mass per unit area
(LMA, for evergreen vs. deciduous). Deciduousness is defined by cold resistance (evergreen vs.
cold deciduous), and drought resistance (evergreen vs. drought deciduous). Grasses are simulated
as tree seedlings with all stems senescent along with leaves at the end of a growing season. The
individuals are reset back to their initial sizes each year and the population density is also reset
by conserving current total biomass. The photosynthesis pathway is predefined as $C_3$ or $C_4$.

We defined 9 PFTs for our test runs in this paper to roughly represent global natural

vegetation functional diversity (Table 1) according to their life form (tree, shrub, and grass),
photosynthesis ($C_3$ and $C_4$), and leaf phenology (evergreen and deciduous). Crop PFTs were not
included because the purpose of this paper is to describe the baseline processes of natural
vegetation and soil biogeochemical cycle. These PFTs have the same physiological and
demographical processes with different parameters (except $C_3$ and $C_4$ photosynthesis pathways)
representing varied strategies in different environments. Thus, for eco-evolutionary and
ecological community assembly simulations, one PFT can switch to another by changing its
parameters for searching competitively optimal plant traits in different environments.
**2.3 Phenology**
The phenology types are defined by two parameters, i.e., a critical low temperature and a critical
soil moisture index, that are used to trigger leaf fall. These two parameters define 4 phenological
types with their possible factorial combinations: evergreen, drought-deciduous, cold-deciduous,
and drought-cold-deciduous. Evergreen PFTs have high resistances to cold (i.e., very low critical
temperature) and drought (very low soil drought). Cold and drought deciduous PFTs have low
critical temperature and soil drought index, respectively. These phenological types represent
different strategies of dealing with environmental stresses and pressure of competition. It is
possible that the evergreen would be more competitive in high seasonality regions (e.g.,
evergreen needle-leaved trees in boreal regions), though the first response of plants to harsh
environments (e.g., cold or dry) is to shed their leaves. Our definition of phenology is designed
to allow the model to evaluate the competitively optimal strategy in future studies.
For the cold-deciduous PFTs (temperate/boreal deciduous broadleaf and cold shrub), we
use the growing degree days above 5 °C ($GDD_5$) to trigger phenological onset and a critical low
temperature ($T_m$) for the offset. $GDD_5$ is calculated from the days that temperature starts to
increase from the coldest days in the non-growing season. The critical GDD for a plant to initiate
growth ($GDD_c$) is defined as a function of chilling days in the non-growing season (Prentice et
al., 1992):

$$GDD_c = a_0 + d \cdot e^{-b \cdot N_{CD}}, \qquad (1)$$

where, $N_{CD}$ is the days of the cold period in nongrowing season before bud burst, $a_0$ is the
minimum $GDD_c$ (50) when the cold period is sufficiently long, $d$ is the maximum addition of
$GDD_c$ (800) when there is no cold period (i.e., $N_{CD}$=0), $b$ is a shape coefficient (0.025). These
parameters are tunable and should change with the acclimation of plants to new climates.
The running mean temperature that represents the mean temperatures over a short period of
time is calculated as:

$$\begin{cases} T_m(i) = T_d(i), & \text{when } i = 1 \\ T_m(i) = 0.8 T_m(i-1) + 0.2 T_d(i), & \text{when } i \geq 2 \end{cases} \tag{2}$$

The critical temperature of triggering leaf senescence ($T_c$) is calculated as a function of the
number of growing days ($N_{GD}$).

$$T_c = T_{0,c} - s \cdot e^{-c \cdot (\max\,(0, N_{GD} - L0))}, \tag{3}$$

where, $T_{0,c}$ is the highest critical temperature when $N_{GD}$ is sufficiently long, $s$ is the range that a
critical temperature can change, c is a shape parameter, $L0$ defines the lowest critical temperature
($T_{0,c}$ -$s$) when $N_{GD}$ is smaller than $L0$. The rationale in this equation is that when a growing
period is not long enough, plants need a lower $T_c$ to trigger leaf fall so that they can have a
growing season that is not too short. This setting is based on the thermal adaptation analysis of
Yuan *et al.* (2011). It balances growing season length and frost risks by adjusting critical $GDD_c$
and $T_c$ according to chilling days and growing days to reduce frost risk in warm regions and
increase growing season length in cold regions. In this way, leaf senescence is also a function of
growing season length and leaf aging. For example, in a region with a longer growing season,
plants will have a higher $T_c$ and initiate senescence when it is still relatively warm.

For the drought deciduous PFTs (tropical drought deciduous broadleaf, arid shrub, and $C_4$

grass), we used a soil moisture index ($s_D$) to start and end a growing season.

$$s_D = \sum_{i=1}^{n} Min\left(1.0, max\left(\frac{\theta_i - \theta_{WP,i}}{\theta_{HC,i} - \theta_{WP,i}}, 0.0\right)\right), \tag{4}$$

where $i$ is the soil layer in the root zone, $\theta$ is soil water content (vol./vol.), $\theta_{WP}$ is wilting point,
and $\theta_{HC}$ is soil water holding capacity. The critical soil moisture values that trigger new leaf
growth and leaf fall are defined as PFT-specific parameters. We slightly tuned these two
parameters according to the soil moistures where the deciduous PFTs' leaves start to grow or
fall. Usually, the critical soil moisture for starting new leaf growth is higher than the soil
moisture that triggers leaf senescence so that the plants can have a stable growing season.
**2.4 Plant allometry and demography**
**Allometry and Plant architecture**
The plant allometry and architecture are critical for plant resources allocation, light capture, and
soil water and nutrients uptake. The allometry equations are the same as those used in LM3-PPA
(Farrior et al., 2013; Weng et al., 2015):

$$\begin{cases} A_C = \alpha_C D^{\theta_C} \\ Z = \alpha_Z D^{\theta_Z} \\ S = 0.25\pi\rho\Lambda\alpha_H D^{2+\theta_H} \,, \\ A_L^* = l_{max}A_C \\ A_{FR}^* = \varphi_{RL}l_{max}A_C \end{cases} \qquad (5)$$

where $D$ is tree diameter; $A_C$ is crown area; $Z$ is plant height; $S$ is woody biomass (sapwood plus
heartwood); $\alpha_C$ and $\alpha_Z$, are the scaling factors for crown area and plant height, respectively; $\theta_C$
and $\theta_Z$ are the exponents for crown area and tree height, respectively; $\pi$ is ratio of a circle's
circumference to its diameter; $\rho$ is wood density (kg C m$^{-3}$); $\Lambda$ is the taper factor from a cylinder
to a tree with the same $D$; $A_L^*$ and $A_{FR}^*$ are the target surface area of leaves and fine roots,
respectively; $\varphi_{RL}$ is the area ratio of leaves to roots. $l_{max}$ is the maximum leaf area per unit crown
area, defined as a function of plant height (Z):

$$l_{max}(Z) = L_{max,0}(Z + h_0)/(Z + H_0), \qquad (6)$$

where $L_{max,0}$ is the maximum crown LAI when a tree is sufficiently tall, $Z$ is tree height, $h_0$ is a
small number that makes a minimum $l_{max}$ when tree height is close to zero, and $H_0$ is a curvature
parameter.

**Plant growth and allocation of carbon and nitrogen to plant tissues**


The allocation of carbon to wood, leaves, and roots is affected by climate and forest age (Litton
et al., 2007; Xia et al., 2019). However, vegetation models cannot capture these patterns well at
large spatial scales, even if the adaptive responses to climate and forest ages are considered (Xia
et al., 2019, 2017), partly because of the absence of explicit representation of shifts in species
composition and competition between individuals (Franklin et al., 2012; Dybzinski et al., 2015).
BiomeE has an optimal growth scheme that drives the allocation of carbon and nitrogen to
leaves, fine roots, and stems based on the optimal use of resources and light competition (Weng
et al., 2019). In this scheme, the growth of new leaves and fine roots follows the growth of
woody biomass (i.e., stems), and the area ratio of fine roots to leaves is kept constant during the
growing season. The allocation of available carbon between structural (e.g., stems) and
functional (e.g., leaves and fine roots) tissues is optimal for light competition at given nitrogen
availability.

Mathematically, differentiating the stem biomass allometry in Eq. 5 with respect to time,

using the fact that $dS/dt$ equals the carbon allocated for wood growth ($G_W$), gives the diameter
growth equation:

$$\frac{dD}{dt} = \frac{G_W}{0.25\pi\Lambda\rho_W\alpha_z(2+\theta_z)D^{1+\theta_Z}} \tag{7}$$

This equation transforms the carbon gain from photosynthesis to the diameter growth that results
from wood allocation and allometry (Eq 5). With an updated tree diameter, we can calculate the
new tree height and crown area using allometry equations, and the targets of leaf and fine root
biomass (Eq. 5). Generally, the growing-season average allocations of carbon and nitrogen to
different tissues are governed by two parameters: the maximum leaf area per unit crown area
($l_{max}$) and fine root area per unit leaf area ($\varphi_{RL}$) (Eq. 5). The optimal-growth allocation scheme
combined with explicit competition for light and soil resources in our model makes it possible to
simulate the underlying processes that determine emergent allocation patterns (Dybzinski et al.,
2011; Farrior et al., 2013; Farrior, 2019; Weng et al., 2019).
**Reproduction and Mortality**
At a yearly time-step, the cumulative carbon and nitrogen allocated for reproduction by a canopy
cohort over the growing season length, $T$, is converted to seedlings according to the initial plant
biomass ($S_0$) and germination and establishment probabilities ($p_g$ and $p_e$, respectively).
Generally, the population dynamics can be described by a variant of the von Foerster equation
(von Foerster, 1959):

$$N(S_0, t) = \frac{p_g p_e}{S_0} \int_0^T N(\tau) G_F(\tau) d\tau$$

$$(8)$$

$$\frac{dN(s,t)}{dt} = -\mu(s,t) N(s,t).$$

where $N(S_0, t)$ is the spatial density of newly generated seedlings, $N(\tau)$ is the spatial density of
this cohort of trees at time $\tau$, $G_F$ is the carbon allocation to seeds, and $\mu$ is PFT-specific mortality
parameter.
Each PFT has a background mortality rate that is assigned from the literature. These
background rates are assumed to be size-independent for the canopy layer trees, but size-
dependent for understory trees. Many factors affect tree mortality, such as light, size,
competition crown damage, hydraulic failure, trunk damage etc. (Lu et al., 2021; Zuleta et al.,
2022). These factors result in high mortality rates of seedlings and old trees (i.e., a "U-shaped"
mortality curve). We use the following equation to delineate a mortality rate that varies with
social status (crown layers), shade effects, and tree sizes:

$$\mu(s,t) = \mu_0(1 + f_L f_S)f_D \tag{9}$$

where $f_L$ is the shade effects on mortality ($f_L = \sqrt{L-1}$), $f_S$ is seedling mortality when a tree is
small ($f_s = A_{SD}e^{-B_{SD}\cdot D}$), and $f_D$ represents the size effect on the mortality of adult trees ($f_D =$
$m_s \frac{e^{A_D(D-D_0)}}{1+e^{A_D(D-D_0)}}$). $L$ is the layer this plant is in ($L=1$ for the canopy layer and 2 for the second,
and so on), $A_{SD}$ is the maximum multiplier of mortality rate for the seedlings in the understory
layers, $B_{SD}$ is the rate of mortality decreasing as tree diameter ($D$) increases, $m_s$ is the maximum
multiplier of mortality rate for large-sized trees, $D_0$ is the diameter at which the mortality rate
increases by $m_s$ /2, and $A_D$ is a shape parameter (i.e., the sensitivity to tree diameter).

## 2.5 Crown self-organization and layering

Tree crowns are arranged into different vertical canopy layers according to tree height and
crown area if their total crown area is greater than the land area following the rules of the PPA
model (Strigul et al., 2008). In PPA, individual tree height is defined as the height at the top of
the crown, and all leaves of a given cohort are assumed to belong to a single canopy layer.
The height of canopy closure for the top layer is referred to as critical height ($Z^*$, the height of
the shortest tree in the layer) and is defined implicitly by the following equation:

$$k(1-\eta) = \sum_i \int_{Z^*}^{\infty} N_i(Z,t)A_{CR,i}(Z^*,Z)dZ \tag{10}$$

where $N_i(Z,t)$ is the density of PFT $i$ trees of height $Z$ per unit ground area; $A_{CR,i}(Z^*,Z)$ is the
crown area of an individual PFT $i$ tree of height $Z$; $\eta$ is the proportion of each canopy layer that
remains open on average due to wind and imperfect spacing between individual tree crowns, and
$k$ is the ground area. The top layer includes the tallest cohorts of trees whose collective crown
area sums to $1-\eta$ times the ground area; lower layers are similarly defined.
All the trees taller than the critical height can get full sunlight and all trees below this
height are shaded by the upper layer trees. Trees within the same layer do not shade each other,
but there is self-shading among the leaves within individual crowns. Cohorts in a sub-canopy
layer are shaded by the leaves of all taller canopy layers. In each canopy layer, all cohorts are
assumed to have the same incident radiation on the top of their crowns. Note, the gap fraction $\eta$
is necessary to allow additional light penetration through each canopy layer for the persistence of
understory trees in monoculture forests in which the upper layer crowns build a physiologically-
optimal number of leaf layers (Farrior et al., 2013). The grasses only form one layer. Those
individuals who cannot stay in that layer because of limited space will be killed (i.e., when the
total grass crown area is larger than the land area).
**2.6 Ecosystem carbon and nitrogen biogeochemical cycles**
There are seven pools in each plant: leaves, fine roots, sapwood, heartwood, fecundity (seeds),
and non-structural carbohydrates and nitrogen (NSC and NSN, respectively). The carbon and
nitrogen in plant pools enter soil pools with the mortality of individual trees and the turnover of
leaves and fine roots. Soil has a mineral nitrogen pool for mineralized nitrogen and five soil
organic matter (SOM) pools for carbon and nitrogen: metabolic litter ($x_1$), structural litter ($x_2$),
microbial ($x_3$), and fast ($x_4$) and slow-turnover ($x_5$) SOM pools.
The decomposition processes are simulated by a model modified from Manzoni et al.
(2010). It was described in Weng et al. (2019, 2017). The decomposition rate of a SOM pool is
determined by the basal turnover rate together with soil temperature and moisture following the
formulation of the CENTURY model (Parton et al., 1988, 1987). The microbial pool transfers
carbon and nitrogen among SOM pools and releases mineralized nitrogen. Microbial carbon use
efficiency (CUE, carbon transfer from litter to microbial matter) is a function of litter nitrogen
content, following the model of Mazoni et al. (2010).

The N mineralization in decomposition is determined by microbial nitrogen demand,

SOM's C:N ratio, and decomposition rate. In the high C:N ratio SOM, microbes must consume
excess carbon to get enough nitrogen for growth. By contrast, in the low C:N ratio SOM,
microbes must release excess nitrogen to get enough carbon for energy. Depending on the C:N
ratios of SOM, soil microbes may be limited by either C or N.

The out-fluxes of C and N from the $i^{th}$ pool ($dC_i$ and $dN_i$, respectively) are calculated by:

$$dC_i = \xi(T, M)\rho_i QC_i,$$
$$dN_i = \xi(T, M)\rho_i QN_i , \qquad (11)$$

where $\xi$ is the response function of decomposition to soil temperature ($T$) and moisture ($M$), $\rho_i$ is
the basal turnover rate of the $i^{th}$ litter pool at reference temperature and moisture, $QC_i$ is the C
content in $i^{th}$ pool, and $QN_i$ is the N content in the $i^{th}$ pool.

The microbial growth ($dM$) is calculated as the co-limit of available carbon and nitrogen

mobilized at this step:

$$dM_i = Min(\varepsilon_0 \cdot dC_i, \Lambda_{microbe} \cdot dN_i), \qquad (12)$$

where $\varepsilon_0$ is default carbon-use efficiency of litter decomposition (0.4) and $\Lambda_{microbe}$ is a microbe's
C:N ratio, which is a fixed value (10 in this model). The soil heterotrophic respiration ($R_h$) is the
microbial respiration (i.e., the difference between carbon consumption and new microbial
growth), and the total N mineralization rate ($N_{mineralized}$) is calculated as the sum of mineralized N
in the SOM pools and microbial turnover:

$$R_h = \sum_{i=3}^{5} dC_i - \sum_{i=4}^{5} M_i,$$

$$N_{mineralized} = \sum_{i=3}^{5} dN_i - \sum_{i=3}^{5} m_i / \Lambda_{\text{microbe}}$$

(13)

The $R_h$ releases to atmosphere as $CO_2$. Mineralized N enters the mineral N pool for plants to use.
The dynamics of the mineral N pool is represented by the following equation:

$$\frac{dN_{mineral}}{dt} = N_{deposition} + N_{mineralized} - U - N_{loss},$$

(14)

where $N_{\text{deposition}}$ is N deposition rate, assumed to be constant over the period of simulation; $N_m$ is
the N mineralization rate of the litter pools (fast and slow SOM and microbes); $U$ is the N uptake
rate (Kg N m$^{-2}$ hour$^{-1}$) of plant roots; and $N_{\text{loss}}$ includes the loss of mineralized N by
denitrification and runoff. The N deposition ($N_{\text{deposition}}$) is the only N input to ecosystems, and we
set nitrogen fixation as zero in this version of the model.

**3 Model test**
For our comparison of model performance against observations and other models, we used
the full demographic version of BiomeE (described above) and also designed a "single-cohort"
version of the model to benchmark our demographic implementations. In the single-cohort
model, the mortality of trees is simulated as the turnover of woody biomass, and the fecundity
resources (carbon and nitrogen) are used to build the same-sized parent trees, instead of
seedlings growing from understory layers. If the total crown area of the trees in this cohort is
greater than the land area, the extra trees will be removed to make the total crown area less than
or equal to the land area. At equilibrium, the turnover of woody biomass is equal to the new
growth each year and the new trees generated from fecundity resources are killed by self-
thinking. The single-cohort model uses the mean state of the canopy layer trees to represent the
characteristics of the whole community. This single-cohort model performs like the traditional
biogeochemical models and simplifies vegetation computation.

In the test runs, the distribution of PFTs was obtained from the Ent vegetation map (Ito et

al., 2020), which was derived from 2004 MODIS land cover and PFT data products (Friedl et al.,
2010) and climate data (Figure 2). For these simulations, croplands and pastures were replaced
by the potential natural vegetation types. We slightly tuned leaf maximum carboxylation rate
($V_{cmax}$) to fit the general pattern of global GPP, while keeping other parameters unchanged.

Forcing data are from the TRENDY project CRU-NCEP data (Sitch et al., 2015) and have

a 6-hour time step at a spatial resolution of 0.5°×0.5°.  These data are available at the website
https://www.uea.ac.uk/web/groups-and-centres/climatic-research-unit/data.
We aggregated these data into 2.0°×2.5° grid cells and used thirty years' of data (1988~2017) to
force the model to run for 600 years, which is long enough for the model to approach equilibrium
states for both vegetation and soil carbon pools. These data include temperature, precipitation,
shortwave radiation, longwave radiation, specific humidity, and wind speed (U and V
directions). We interpolated the radiation data ($R_S$) into half-hour timesteps based on the sun
zenith angle ($\theta_S$) and radiation penetration rate calculated from data.

$$R_S(t) = \left( \frac{R_{H6}}{S^* cos cos\, \theta_S(H6)} \right) S^* cos\, cos\, \theta_S(t) \; , \tag{15}$$

where $S^*$ is solar constant (1362 W/m$^2$).  Other variables are linearly interpolated to the model
run time step, half hour in this study. Atmospheric $CO_2$ concentration is set at the model default
level (350 ppm).

**3.1 Data sources for model evaluation**

**The LAI data** were from the Ent vegetation dataset (Ito et al., 2020), where the LAI was derived from 2004 MODIS LAI data (Tian et al., 2003, 2002). **Gross primary productivity (GPP) data** are from a global retrieval of GPP using remote sensing observations. These data are on a 1°×1° geographic grid at a monthly time step based on an Artificial Neural Network retrieval algorithm (Alemohammad et al., 2017). This algorithm uses six remotely sensed observations as input: Solar Induced Fluorescence (SIF), Air Temperature, Precipitation, Net Radiation, Soil Moisture, and Snow Water Equivalent. The data are available from 2007 to 2015. **The tree height data** are from spaceborne light detection and ranging (lidar) global map of canopy height at 1-km spatial resolution developed by Simard et al. (2011). These authors used the 2005 data from the Geoscience Laser Altimeter System (GLAS) aboard ICESat (Ice, Cloud, and land Elevation Satellite) to derive global forest canopy heights. **Biomass data** are from a Global 1-degree Maps of Forest Area, Carbon Stocks, and Biomass, 1950-2010 developed by Hengeveld et al. (2015). **Soil carbon data** are from Food and Agriculture Organization (FAO) Harmonized World Soil Database (version 1.2), updated by Wieder et al. (2014).

**MsTMIP model simulation data**

We selected six model simulations (BiomeBGC, CTEM, CLM4, LPJ, ORCHIDEE, VEGAS) from the Multi-scale Synthesis and Terrestrial Model Intercomparison Project (MsTMIP) (Huntzinger et al., 2013) to compare against our model simulations. These models are well-developed and widely used in Earth system models, representing the state-of-art of current land vegetation model development. MsTMIP provided prescribed land use types for all the participant models. However, it is up to the participant models to simulate disturbance impacts on ecosystems (Huntzinger et al., 2013). MsTMIP conducted five sets of experimental runs with

different climate forcing, land-use history, atmospheric $CO_2$ concentration, and nitrogen
deposition. In this study, we compared to the SG1 simulation experiment because it is driven by
the 1901~2010 climate forcing data with constant $CO_2$ concentration and constant land cover
(Huntzinger et al., 2013), which are the closest to our model runs.

**3.2 Selected Grid Cells for Comparison**

To illustrate model behavior, we selected 8 grid cells that cover boreal forests, temperate forests,
tropical forests, $C_4$ grasslands, and arid shrublands to show the simulated ecosystem
development patterns across the climate zones with different dominant PFTs (Table 2). Brazil
Tapajos (TPJ), Oak Ridge (OKR), Harvard Forest (HF), Manitoba old black spruce site (MNT),
and Bonanza Creek (BNC) are covered by tree PFTs. Konza long-term ecological research
station (LTER) (KZ) is $C_4$ grass. Walnut Gulch Kendall (WGK) and Sevilleta LTER (SV) are
covered by arid shrubs. These sites were chosen because they have extensive data on vegetation
and climate conditions for future comparisons.

**4 Results**

**4.1 Simulated vegetation structural and ecosystem carbon dynamics**

In the forest sites, the simulated vegetation structure by the full demographic model changes with
the growth, regeneration, and mortality processes (Figure 3). The temporal dynamics of the
canopy development can be separated into three stages according to the canopy crown dynamics:
1) open forest stage, 2) self-thinning stage, and 3) stabilizing stage.

In the open forest stage, the crown area index (CAI) is less than 1.0 and all the individuals are in full sunlight. The tree crowns grow rapidly to occupy the open space (Figure 3: a). In the self-thinning stage, the open space is filled by the crowns of similar sized trees (i.e., the forest is closed) and canopy trees are continuously pushed to the lower layer(s) (i.e., self-thinning) and the CAI continues to increase due to the limited space with growing tree crowns (i.e., the new spaces vacated from the canopy tree mortality cannot meet the space demand from crown growth). The sizes of trees in the canopy layer are still similar (Figure 3: b, c) and the critical height (the shortest tree height in top layer) keeps increasing in this period.

In the stabilizing stage, when the space generated by the mortality of canopy trees is larger than the growth of canopy tree crown area, no trees are pushed to the lower layer and the lower layer trees start to enter the canopy layer, leading to a sharp decrease in critical height (Figure 3: b) and the mixing of different sized trees in the canopy layer. The CAI is decreasing as well because of the high mortality rates of the understory layer trees. The growth, regeneration, mortality, and space filling processes are eventually equilibrated with model run, and the forest structure is then stabilized.

The tallest plant height (Figure 3: c), the height the tallest cohort, keeps increasing as this cohort exists. The sharp decrease indicates a replacement by or merging with another shorter cohort because the density of trees in this cohort is low (0.0001/ha in this case) or the similarity between the tallest and the second tallest is high. The total basal area (Figure 3: d) is an index of the sum of all trees at a site. It keeps increasing during forest development and is equilibrated earlier than height and crown structure.

In these sites, at equilibrium, the tropical forest site (TPJ) has the highest crown area index (around 2.2), followed by warm temperate forest at OKR, mixed forest at HF, and boreal forests

at BNC and MNT (Figure 3). The shrubs and grasslands in arid regions have the lowest crown
area index (CAI), with basal area following similar patterns. For forested sites, tree height is
tallest at TPJ, followed by OKR, HF, MNT, and BNC. The shrubs are short according to their
allometry parameters and the height of grasses during non-growing season is zero. The critical
height, which separates canopy layer trees from the understory layers, follows the same order as
that of tree height with high fluctuations with cohort changes. (More cohort details are in
Supplementary Information Figures S1-S8)

For the temporal dynamics in the full demographic simulation (Figure 4), the simulated

GPP aligns closely with LAI and they reach their equilibrium states at similar times across sites
(Figure 4: a, b). According to the definition of maximum crown LAI ($l_{max}$) in Eq. 6, the grass
LAI (i.e., Konza) reaches the maximum each year, except the first year due to the low initial
density (Figure 4: a). The biomass accumulation is much slower in forests because of the longer
time needed for forest structure (size distribution) to reach equilibrium. Soil carbon equilibration
is faster in the warm regions than in cold regions overall because of the higher turnover rate of
SOM pools in warm regions. At equilibrium, forested sites have higher LAI, biomass, and
carbon stocks per area compared to the shrub and grass sites overall. Vegetation biomass is
lowest at the grassland site, Konza LTER, because, within the model, the grassland ecosystems
cannot accumulate persistent biomass.
The PFTs at TPJ and MNT are evergreen trees. Their LAI does not change over the whole
year (Figure 5: a). The forest in OKR has the longest growing season in the three deciduous
forest grids, followed by HF and BNC. BNC's growing season is only around 120 days, about
half of OKR's growing season. The growing season of grasses in KZ starts in late May and ends
in September. The two arid-adapted shrub sites (SV and WGK) are controlled by water
availability. In TPJ (tropical evergreen forest), the trees have photosynthesis throughout the
entire year (Figure 5: b). In MNT, photosynthesis only happens in warm seasons with the leaves
kept in the crowns because of the dominant PFT is evergreen needleleaf tree. The deciduous
trees in OKR and HF have high photosynthesis rates during the growing season. The
photosynthesis rates in SV and WGK are generally low because of the dry environments.
However, the precipitation events can drive photosynthesis rates high in these arid regions.
As shown in Figure 6: a, the evergreen needle-leaved forests keep their leaves in northern
high latitude regions during January, while the photosynthesis rate in this region is low (Figure 6:
b). In July, northern high latitude regions green up and their photosynthesis rates are high in wet
regions. The single cohort model run predicts a similar pattern because of the same phenology
model (Figure S9).

**4.2 Global Comparisons with Observations**
The simulated LAI roughly capture the spatial pattern of MODIS LAI (Figure 7: a and b), though
there are high variations at each grid (Figure 8: a). Generally, the simulated LAI in well
vegetated grids, e.g., boreal forest regions, is underestimated by our model because the crown
LAI is calculated as a function of tree height and a parameter of maximum crown LAI (Table 1
and Eq. 6). The LAI in the grids that were converted to different land use types is overestimated
because we assume all terrestrial grids are covered by potential vegetation in our test runs.
Compared with the SIF GPP (Alemohammad et al., 2017), simulated GPP is higher than
the SIF GPP generally, though lower in arid regions (Figure 7: c, d and Figure 8: b). The
simulated tree height (Figure 7: e, f and Figure 8: c) is mostly taller compared to observations
(Simard et al., 2011) because most forests have been altered by human activities (Pan et al.,
2013). However, the simulations and observations cover approximately the same range of tree
heights (up to 40 m). Simulated biomass is much higher than the observations (Figure 7: g, h and
Figure 8: d) because, in the observations, many forest regions have been transformed to low
biomass land use types (such as croplands) or represent earlier successional stages with less
accumulated carbon (i.e., not equilibrium states).
Simulated soil carbon does track the observations (Figure 7: i, j and Figure 8: e) better than
biomass, likely because soil carbon stocks are more stable compared to biomass in response to
disturbances and human activities. For areas where the model underpredicts soil carbon, the
difference could arise from the missing biogeochemical processes that may lead to high carbon
accumulation in some regions (e.g., peats) (Davidson and Janssens, 2006; Briones et al., 2014;
Euskirchen et al., 2014) and the relatively high uncertainties in the soil carbon data (Tifafi et al.,

2018).


## 4.3 Comparison with MsTMIP models

We compared the performance of our model with MsTMIP models at the 8 locations that were
used to show ecosystem development patterns (Table 2). For most of these sites, LAI in BiomeE
is lower compared the other MsTMIP models (Figure 9: a), while the estimated GPP is within
the range of MsTMIP predictions (Figure 9: b). LAI differences are a consequence of the
formulations within BiomeE, as described further in the Discussion (5.2 Model predictions and
performance). Specifically, BiomeE simulates leaf growth by using a maximum crown LAI,
which is lower than the real forest LAI.
The low LAI does not affect crown total photosynthesis because leaves in lower canopy
layers contribute little to the total carbon assimilation. BiomeE predicted biomass (Figure 9: c)
and soil carbon (Figure 9: d) generally fall towards the higher end of the MsTMIP simulations,
except for the more arid grass- and shrub-dominated sites. We note, however, that there are wide
differences in estimates for vegetation and soil carbon across the models, likely because of
different treatments of mortality and decomposition functions in these models.
More broadly, the latitudinal mean of BiomeE simulated GPP is at the lower end of
MsTMIP model predictions (Figure 10: a). Since BiomeE's GPP was tuned to fit remote sensing
data derived GPP, the MsTMIP models may over-estimate global GPP. The net primary
production (NPP) (Figure 10: b), plant carbon (Figure 10: c), and soil carbon (Figure 10: d)
simulated by BiomeE are within the range simulated by the MsTMIP models. This indicates that
BiomeE has slightly lower respiration than the MsTMIP models. In the arid regions (e.g., around
latitude 40-50 $^{o}$S of South America), we simulated a lower GPP than that of MsTMIP models
because of high drought sensitivity in our model.
The demographic processes have significant impacts on the simulated GPP, biomass, soil carbon,
and vegetation structure compared to the single-cohort BiomeE (Figure 11). The full
demographic BiomeE includes an understory layer of plants, resulting in higher LAI in high LAI
regions and also slightly higher GPP. However, the total biomass predicted by the two model
settings are similar because of the tradeoffs in allocation between leaves and stem growth and
tree size distribution and because most biomass is in woody tissues (see Figures S10 and S11 in
the Supplementary Information for the single cohort BiomeE simulations).  In the full
demography model, tree mortality removes all the biomass, including leaves, fine roots, and
stems, while in the single-cohort model, the mortality is represented as the turnover of woody
biomass. Consequently, the full demography model has higher emergent turnover rate for the
whole vegetation carbon pool.

Compared to the single-cohort model, the full demography model predicts higher LAI and

GPP in warm and wet regions and lower LAI and GPP in cold and dry regions (Figure 12: a, b).
The full demography model also predicts much lower biomass and soil carbon than the single-
cohort model in cold and dry regions (Figure 12: c). The reduced biomass input from full
demography alone is causing the difference in SOM dynamics since the two models share the
same SOM pools and turnover/decomposition processes. Demographic processes greatly reduce
model stability because low reproduction and high mortality rates in dry and cold regions can
greatly reduce vegetation coverage. By contrast, the single-cohort model replaces these processes
by simplified turnover of plant carbon pools that allows plants to stay in extremely dry or cold
conditions.

**4.4 Eco-evolutionary simulation and sensitivity test**
The BiomeE model has the potential to predict competitively dominant PFTs in the continuum of
plant traits through game-theoretic simulations according to the principles of evolutionarily
optimal competition. We illustrate this with a set of simulations conducted at a series of
ecosystem nitrogen content (from 269 to 575 g N/m$^2$) with five PFTs sampled from the
continuums of LMA ($\sigma$, from 0.06 to 0.14) and target root/leaf area ratio ($\varphi_{RL}$, from 0.8 to 1.2
corresponding to each LMA). The simulations were set as nitrogen-closed (i.e., no input and
output of nitrogen). The differences in ecosystem total nitrogen represent the environmental
conditions that arise from soil and climate conditions. At the lowest ecosystem total nitrogen
(Figure 13: a), the PFT with highest LMA (0.14 kg C/m$^2$ leaf) wins. As the ecosystem total
nitrogen increases (Figure 13: b - d), the winner shifts from high to low LMA PFTs. This means
that in infertile soils or cold climates where biogeochemical cycles are slow (e.g., tundra and
boreal forests), the eco-evolutionarily optimal PFTs should have high LMA leaves, and vice
versa. This pattern is consistent with the predictions of a theoretical model in Weng et al. (2017).
This simulation is also a case of the sensitivity test of vegetation dynamics at different
environmental conditions. Vegetation can shift their compositions and dominant plant traits to
maintain an eco-evolutionarily optimal state, and thus amplify or attenuate the responses of
ecosystem carbon cycle to climate changes.
**5 Discussion**
We developed a parsimonious terrestrial ecosystem model for ModelE to simulate vegetation
dynamics and ecosystem biogeochemical cycles. This model includes a cohort-based
representation of vegetation structure, a height structured light competition scheme, demographic
processes, and coupled carbon-nitrogen biogeochemical cycles. This model has four major
modules that organize the hierarchical processes of ecosystems together into a cohesive
modeling structure: 1) plant physiology (i.e., photosynthesis and respiration), 2) plant phenology
and growth, 3) vegetation structural dynamics, and 4) soil biogeochemical cycles (Figure 1).
Each module is cohesive and has a minimum set of variables as the input from other modules.

## 5.1 Model formulation


In designing this model, we considered the simulation of competitively optimal strategy of plants
in different climates based on fundamental ecological rules (Purves and Pacala, 2008; Falster and
Westoby, 2003; Franklin et al., 2020). These strategies are mainly related to light competition,
water conditions, nutrient use efficiency, and disturbances (e.g., fire), and represented by the
traits of wood density, height growth, leaf longevity, and photosynthesis pathways. PFTs are
used in this model as an integrative unit representing combinations of plant traits for simulating
(1) the spontaneous dynamics of carbon, water, and energy fluxes as the core functions of an
ESM-based land model and (2) the transient vegetation structural and compositional dynamics
and ecosystem biogeochemical cycles in response to climate variations.
We adopte a generic design of the PFTs by defining them as samples from the high
dimensional space defined by plant traits in their natural ranges. This approach substantially
simplifies the parameterization of PFTs because it becomes the selection of strategies in different
trait values (i.e., parameters). The numbers of PFTs are flexible, depending on what strategies
the users wish to simulate (as the test simulations in Figure 13). Thus, the PFTs are adaptive and
variable in different environmental conditions, making it possible to reduce the number of PFTs
while representing functional diversity and the optimal adaptation to climate conditions.
To represent the major variations in plant functional diversity, we chose four plant traits as
the primary axes to define PFTs: wood density, LMA, height growth parameter, and leaf
maximum carboxylation rate. Wood density is relatively conservative (Swenson and Enquist,
2007; Chave et al., 2009), mostly ranging from 200 to 500 kg C m$^{-3}$, while herbaceous stem
density ranges from 400~600 kg C m$^{-3}$ (Niklas, 1995). However, herbaceous stems are usually
hollow, making the ratio of total biomass to its volume low, and grasses shed their stems each
growing season, resulting in faster stem turnover. It is a strategic difference from woody plants,
which keep the woody tissues to build up their trunks and thus display their leaves on top of
trunks for light competition (Dieckmann et al., 2007; Falster and Westoby, 2003). LMA is the
key leaf trait that determines leaf life longevity and leaf types (i.e., evergreen vs. deciduous) )
(Osnas et al., 2013), and represents the strategy for the competition in different soil nutrient
levels (Tilman, 1988; Reich, 2014; Weng et al., 2017) and resistance to stresses of water and
temperature (Oliveira et al., 2021).
The phenological type is simulated as an emergent property of plant physiological
processes and strategies of dealing with seasonal air temperature and soil water variations. Three
parameters – growing degree days, running mean daily temperature, and critical soil moisture –
are used to define all possible phenological types. These three parameters are widely used in
phenology modeling (e.g., Sitch et al., 2003; Prentice et al., 1992; Arora and Boer, 2005).
However, phenology is not just a physiological response to the seasonality of climate conditions.
Evergreen plants are distributed in periodically cold or dry climates. It is a competitively optimal
strategy in infertile soil conditions (Aerts, 1995; Givnish, 2002; Coomes et al., 2005). The
benefits and costs of keeping different leaves in cold or dry periods should be realistically
simulated based on eco-evolutionary theories for phenology modeling (e.g., Levine et al., 2022;
Weng et al., 2017).
As for soil organic matter decomposition, the CASA model, which has 13 pools with
different transfer coefficients and turnover rates (Randerson et al., 1997; Potter et al., 1993,
2003), is currently used in ModelE. The soil biogeochemical cycle models developed thereafter
have more sophisticated processes, especially those of microbial activities and carbon use
efficiency (Manzoni et al., 2010; Wieder et al., 2014; Wang and Goll, 2021), and simplified
carbon pools, mostly following the CENTURY model structure (Parton et al., 1987). We chose
an intermediate complexity scheme that has only two SOM pools but a functional microbial pool
for decomposing SOM (Manzoni et al., 2010; Weng et al., 2017) so that the dynamics of SOM's
C:N ratio, carbon use efficiency, and nitrogen mineralization can be reasonably simulated while
keeping the model structure parsimonious.

## 5.2 Model predictions and performance

We only evaluated the carbon cycle in the model simulations in this paper, though the
nitrogen cycle is also simulated in tandem with the carbon cycle in the model. The major
processes of this model, e.g., photosynthesis, respiration, phenology, growth, allocation,
demography, soil biogeochemical cycles, are from well-developed models and have been shown
able to capture observational patterns. Data assimilation approaches can be implemented when
parameter tuning becomes essential (Luo et al., 2011; MacBean et al., 2016). So, we did not
extensively tune model parameters to fit observations because the purpose of this paper is to
describe the formulation of the model.
The simulations demonstrate that this model can capture the global patterns of LAI, GPP,
tree height, biomass, and soil carbon (Figure 7), even though the parameters are not extensively
tuned. For example, global GPP patterns are consistent with those derived from SIF data
(Figure7: c, d and Figure 8: b), and simulated tree heights span the same ranges of those derived
from data. The simulated LAI is segregated by PFTs (Figure 8: a), largely because of the
different parameter values of the maximum crown LAI for each PFT. The simulated biomass and
soil carbon is generally higher than those of observations, though simulated soil carbon is lower
in some cold regions.
Several factors likely explain the apparent discrepancies between simulated and observed
LAI, GPP, biomass, and soil carbon. First, the model uses a potential PFT distribution and does
not account for land cover change and land use history. For example, carbon dense ecosystems
(e.g., forests) have been extensively replaced by croplands and pastures. Second, while
vegetation in the real world reflects a variety of successional stages and the effect of various
disturbance events, our model analyses are based on equilibrium simulations without explicit
disturbances, such as fire, deforestation and regrowth. Third, the model assumes mineral nitrogen
is saturated and can consistently meet demands for plant growth. We did not fix the land cover
mismatches by compromising ecosystem physiological processes because we cannot put all these
effects into current model structure (i.e., mortality) when many processes are missing.
LAI is an illustrative variable for understanding why compromises are necessary when
integrating ecological and demographic processes into an ESM. As a critical prognostic variable
in vegetation models, it links both plant physiology and biogeophysical interactions with climate
systems (Richardson et al., 2012; Kelley et al., 2020; Park and Jeong, 2021). While LAI is
usually simulated by a fixed allocation scheme, even if the allocation ratios are dynamic with
vegetation productivity or environmental conditions (Montané et al., 2017; Xia et al., 2019), the
prediction of LAI is often simplified as the balance between leaf growth and turnover.
In practice, modelers tend to tune the LAI to fit observations and get the required albedo
and water fluxes whatever the parameters of photosynthesis and respirations are. The uniform
leaves within a crown would make the lower layer leaves have a negative carbon gain if the LAI
was tuned close to that observed in tropical and boreal evergreen forests (around 5~7).
Therefore, the photosynthesis rate must be tuned to fit the canopy photosynthesis by keeping
these carbon negative leaves.  The crown with carbon negative leaves do not affect the
ecosystem carbon dynamics in the "single-cohort" models because the whole canopy net carbon
gain can be tuned to fit the observations. However, in demographic models, different-sized trees
are explicitly represented and placed in specified crown layers. If the LAI is high, the vegetation
community can create a dark understory where the seedlings cannot survive because of the
negative carbon gain (Weng et al., 2015).
Since the leaf traits in the crown profile are functions of light, water and nitrogen
(Niinemets et al., 2015), a more complex crown development module is required to simulate
branching and leaf development and deployment processes. Plants can optimize canopy leaf
profile to maximize their fitness as a result of interactions among crown structure, light
interception, and community-level competition (Anten, 2002; Hikosaka, 2005; Niinemets and
Anten, 2009; Hikosaka and Anten, 2012).  For balancing the model complexity and computing
efficiency, we defined a low target LAI in this model to avoid carbon negative leaves.
The parameter $V_{cmax}$ used in this model is also much lower than measured in young leaves
(Bonan et al., 2011). The mean photosynthetic capacity of the leaves in a crown is affected the
aging of leaves and their light environment (Niinemets, 2007; Kitajima et al., 2002; Hikosaka,
2005). The new leaves that are usually measured have much higher $V_{cmax}$ than the mean of
canopy. If the leaves were not specifically chosen, the mean of measured $V_{cmax}$ is much lower
than those used in models as shown in Verryckt et al. (2022). This also indicates that $V_{cmax}$ in
current vegetation models is over-estimated.
In this model, the formulation of allometry makes the whole-tree's photosynthesis and
respiration proportional to crown area, and thus the growth rate of tree diameter independent of
crown area. The allocation scheme between the growth of stems and functional tissues (i.e.,
leaves and fine roots) is the strategy of resources foraging for light and soil resources, including
height-structured competition for light. The vital rates drive vegetation structural changes and
biogeochemical cycles (Purves et al., 2008). Our model allows the simulation of vegetation
composition and structural dynamics based on the fundamental principles of ecology, and the
transient changes in terrestrial ecosystems in response to climate change. This model therefore
has the potential to predict competitively dominant strategies represented by plastic plant traits
(e.g., the competitively dominant LMA in the simulations of Figure 13), and the vegetation
structure and compositions that can be eco-evolutionarily optimized.

**5.3 Major uncertainties in BiomeE**
Global vegetation models typically require simplifying assumptions to organize ecosystem
processes at different scales into a cohesive model structure that balances the complexity of
ecosystem processes and the limitations of our knowledge (Prentice et al., 1992, 2007; Harrison
et al., 2021). In our model, many processes, including phenology and drought effects, are based
on phenomenological equations representing the poorly understood links between processes
needed by the model to simulate the entire system. In the following sections, we highlight these
assumptions and evaluate their relative benefits and costs. Transparency in the description of a
community model such as this one will help future developers understand model compromises
and the processes that should be improved. The following phenomenological relationships
represent the major sources of uncertainty in this model.

Water limitation of photosynthesis is calculated as a function of relative soil moisture

following the water stress function from Rodriguez-Iturbe et al. (1999):

$$\beta_D = Min\left(1.0, max\left(\frac{s_D - s_{min}}{s^* - s_{min}}, 0.0\right)\right), \tag{16}$$

The parameters $s^*$ and $s_{min}$ are PFT-specific, representing different responses of PFTs to soil
water conditions, and $S_D$ is the relative soil moisture ranging from 0 (soil water content at wilting
point) to 1 (at field capacity). This formulation that scales soil moisture to a scalar between zero
to 1 is repeatedly used in both physiological responses of photosynthesis and phenology in
ecosystem models as a simplistic treatment of the central role of water limitation on plant
physiology (Powell et al., 2013; De Kauwe et al., 2015; Harper et al., 2021). This equation does
not include the detailed processes of plant hydraulics and its adaptation to arid environments.

Multiple processes are involved to deal with water stress, such as regulating stomata

conductance, shedding leaves, producing more roots, etc. (Oliveira et al., 2021; Volaire, 2018).
On top of these underlying processes, competition and evolutionary processes filter community
emergent properties (Franklin et al., 2020; van der Molen et al., 2011). For example, trees in
different climate regions have similar hydraulic safety margins (Choat et al., 2012), partly due to
the intense competition for light (height growth) and water (root allocation) that require optimal
use of available resources at any climate conditions (Gleason et al., 2017; Liu et al., 2019).
However, in this model, the drought responses are only delineated by Eq. 16. The parameter
choices for $s^*$ and $s_{min}$ likely explain the amplified water stresses and low productivity in arid
regions within our model.

Phenology represents the seasonal rhythms of plant physiological activities as adapted to

periodic changes in temperature, precipitation, and light availability (Abramoff and Finzi, 2015;
Caldararu et al., 2014; Chuine, 2010). DGVMs normally simulate leaf onset and senescence
based on temperature conditions for cold deciduous plants and soil water conditions for drought
deciduous plants (Arora and Boer, 2005; Caldararu et al., 2014). Phenology modeling is still
highly empirical, although new models and approaches for cold deciduous and drought
deciduous strategies have been proposed recently (e.g., Caldararu et al., 2014; Dahlin et al.,
2015; Manzoni et al., 2015; Chen et al., 2016). We used a simple formulation of temperature and
drought responses (Eqs. 1 and 3). These relationships are phenomenological. Future model
development should incorporate eco-evolutionary mechanisms that are selected in the evolution
history.
Mortality is an integrative process of accumulative physiological stresses, structural
damages, and disturbances in a tree's lifetime. The direct causes can be starvation, structural
failure, hydraulic failure, etc. (McDowell, 2011; Aakala et al., 2012; Aleixo et al., 2019). We
only consider the background mortality and define its rate as a function of tree diameter and light
environment (Eq. 10). Hydraulic failure-induced mortality is required for realistically modeling
plant responses to climate changes.
We used these general phenomenological equations primarily because of our knowledge
gaps in ecosystem ecology. We are using the key variables that characterize ecosystem properties
to define the basic model structure but have to use less-than-solid information to link them
together by phenomenological relationships, as all the models do. In addition, our interest is to
keep this model as simple as possible to improve interpretability and transparency and to reduce
the computational burden when it is integrated into the ModelE.  In these places where the
tradeoff between model complexity and process accuracy is necessary, we highlight the
underlying assumptions clearly, rather than implementing temporary fixes that lack solid
empirical evidence.

**5.4 Insights from comparison with MsTMIP models**

Most MsTMIP participant models used in this study have been analyzed by a model traceability
method developed by Xia et al. (2013), which hierarchically decomposes model behavior into
some fundamental processes of ecosystem carbon dynamics, such as GPP, CUE, allocation
coefficients, carbon residence time, carbon storage capacity, and environmental response
functions (Xia et al., 2013; Cui et al., 2019; Zhou et al., 2021). This method is based on the
assumptions of the linear system and the ecosystem emergent behavior per se (Eriksson, 1971;
Emanuel and Killough, 1984; Luo et al., 2012; Sierra et al., 2018), making it is consistent with
the concepts that are used as the basis of ecosystem carbon cycle models. The analyses of model
traceability found, for the carbon cycle dynamics, the major uncertainty is from the modeling of
the turnover rates (reciprocals of residence time) of vegetation and soil carbon pools (Chen et al.,
2015; Jiang et al., 2017). From CMIP5 to CMIP6, the modeling of NPP has been greatly
improved, while the ecosystem carbon residence time remains highly biased (Wei et al., 2022).
According to the traceability analysis approach (Xia et al., 2013), BiomeE also has a high
uncertainty in the modeling of residence times of vegetation and soil carbon pools, because the
mortality is picked up from the global forest data and the SOC decomposition processes are
highly simplified. These issues have been discussed in the section of "5.3 Major uncertainties in
BiomeE". These concepts (e.g., residence time, allocation coefficients) describe model emergent
properties resulting from the underlying biological and ecological processes (i.e., micro-
dynamics vs. macro-states). Fitting the emergent properties directly to improve model behavior
is natural and convenient because many vegetation models are using these emergent properties
(e.g., CUE, residence time, and allocation coefficients) to describe ecosystem processes in their
formulations as a tradition of ecosystem modeling.

There are some common and long-lasting issues in terrestrial ecosystem modeling, such as

responses to warming, responses to atmospheric $CO_2$, drought stress effects, and vegetation
compositional changes (Luo, 2007; Franklin et al., 2020; Harrison et al., 2021). These issues
represent our knowledge gaps in ecosystem ecology. For modeling vegetation dynamics eco-
evolutionarily, we need to use the fundamental ecological processes and unbreakable physical
rules to simulate the emergent processes (e.g., Scheiter et al., 2013; Weng et al., 2019), With the
design of vegetation modeling in the BiomeE, such as the explicit demographic processes,
individual-based competition for different resources, and flexible trait combinations of PFTs, this
model is able to predict some key emergent dynamics of ecosystems based on the underlying
biological and evolutionary mechanisms (as shown in Figure 13). Data from field experiments
(Ainsworth and Long, 2004; Crowther et al., 2016), observatory networks (e.g., Fluxnet,
Baldocchi et al., 2001; Friend et al., 2007), and remote sensing (Duncanson et al., 2020), can
provide direct information for modeling the underlying ecological processes and for validating
predicted emergent properties.

**5.5 Model stability and complexity**

Ecosystem demographic processes (e.g., reproduction and mortality) are a source of high

sensitivity and uncertainty in BiomeE. In some environmental conditions, especially in dry or
cold regions, the predefined parameters can lead to high mortality or failure of reproduction,
making the ecosystems highly instable. To understand these issues, we used the "single-cohort"
version of the model to aid in the diagnosis of issues in the full demographic version of the
model. The major issue we identified is that the model formulation is based on functional
processes in highly productive regions, whereas the model is applied globally and across much
more diverse environmental conditions (e.g., arid environments). The variables and parameters
that work well in highly productive regions (e.g., initial seedling sizes, default leaf growth,
minimum allocation ratios, etc.) are often unsuitable in regions with high environmental stresses.
Although plants have evolved special features to deal with extreme conditions (Lloret et al.,
2012; Reyer et al., 2013; Singh et al., 2020), these features have not yet been well represented in
ecosystem models.

There is a tendency in current DGVMs to use plant physiological trait changes as a

surrogate of community compositional shifts. This approach is usually characterized as
"parameter dynamics" or "response functions" (Fisher and Koven, 2020; Luo and Schuur, 2020)
for reducing model processes and complexity. Adding new processes to work around existing
problems, instead of redesigning the fundamental model processes, is common in model
development. It is helpful for tracking model development, undoing wrong additions, and
improving model performance. However, work-arounds often increase model complexity
without concomitant improvements in model predictions.

Generally, a model's traceability can be improved by transparent assumptions, a well-

defined model structure, and testable output (Famiglietti et al., 2021; Forster, 2017; Hourdin et
al., 2017). Data assimilation approaches improve model parameterization more efficiently and
effectively than manually tuning individual parameters (Wang et al., 2009; Williams et al., 2009;
MacBean et al., 2016) and allow for more detailed uncertainty analysis (Luo et al., 2009; Weng
et al., 2011; Weng and Luo, 2011; Xu et al., 2006; Dietze, 2014). It is important to only include
necessary assumptions in a model and to include them in a way that does not compromise other
processes or parameters. Additionally, many specifications of model formulation are based on
the questions of specific research. We should not expect to develop an all-encompassing model
that fits all application scenarios. On the contrary, maintaining model flexibility and transparency
is critical for using this model as a tool to explore specific science questions. In BiomeE, we
have opted for what we consider the most parsimonious and, at the same time, theoretically
sound formulations of ecosystem processes to allow for computational efficiency in capturing
vegetation dynamics and ecological principles in the context of an ESM.

**5.6 Legacy limitations of ModelE coding and development conventions**
The legacy of model structure and the history of model development can greatly affect the
functions and the selection of model formulations (Alexander and Easterbrook, 2015). ModelE
was developed as a general circulation model, and vegetation in the model to date has been
represented with a set of static biophysics parameterizations to regulate exchanges of energy and
moisture between the land surface and the atmosphere (Hansen et al., 2007; Schmidt et al., 2014;
Kelley et al., 2020). To advance the functionality of the vegetation and the land surface model
within ModelE, increases in complexity must therefore be balanced with the computational
demands of the fully coupled model.
In ModelE, the land model, TerraE, is used to calculate land surface (including vegetation)
water and energy fluxes and soil water dynamics based on the characteristics of vegetation
derived from the vegetation model (e.g., canopy conductance, wetness, etc.) at the grid scale. It
does not calculate each cohort's transpiration and water uptake. In BiomeE, the water limitation
of stomatal conductance is calculated as a function of soil water stress index and root vertical
distribution, instead of the direct plant root water supply (plant hydraulics). This setting works
well for the big leaf model (one canopy at one grid). However, when multiple cohorts of plants
are represented, as we do in BiomeE, it is unable to represent water competition and differentiate
the contribution of each single cohort's contribution to the total transpiration. A structural change
will be required to solve this problem by calculating transpiration from the bottom-up (i.e., from
cohort up to grid cell).

**6 Conclusions**
We developed a demographic vegetation model to improve the representation of terrestrial
vegetation dynamics and ecosystem biogeochemical cycles in the NASA GISS Earth system
model, ModelE. This model includes the processes of plant growth, mortality, reproduction,
vegetation structural dynamics, and soil carbon and nitrogen cycling. To scale this model
globally, we added a new set of plant functional types to represent global vegetation functional
diversity and introduced new phenology algorithms to deal with the seasonality of temperature
and soil water availability. Competition for light and soil resources is individual-based, which
makes the modeling of eco-evolutionary optimality possible. This model predicts the dynamics
of vegetation and soil biogeochemistry including leaf area index, vegetation structure (e.g.,
height, tree density, size distribution, crown organization), and ecosystem carbon and nitrogen
storage and fluxes. This model will enable ModelE to simulate long-term biogeophysical and
biogeochemical feedbacks between the climate system and land ecosystems at decadal to century
temporal scales. It will also allow for the prediction of transient vegetation dynamics and eco-
evolutionary community assemblage in response to future climate changes.

**Code and data availability**
Model codes used in this study (including ModelE2.1, BiomeE module, and the standalone
BiomeE) and the simulations and validation data have been archived at Zenodo
(https://doi.org/10.5281/zenodo.7125963). The updates of model codes will be released with new
versions of GISS ModelE (https://www.giss.nasa.gov/tools/modelE/). The latest standalone
BiomeE is available at GitHub (https://github.com/wengensheng/BiomeESS).

**Author contributions**
EW coded the model and performed test runs and data analysis. EW and BIC wrote the first draft
of the manuscript. BIC, MJP, SSM, NYK, and EW designed the functional coupling with
ModelE and the land module. NYK, IA, RS, and MK contributed to input data, the IO structure
and the coupling between BiomeE and ModelE.  KW, RD, CE, and SWP contributed to
conceptual model development and PFT design. All co-authors contributed to writing or
improving the manuscript.

**Competing interests**
The authors declare that they have no conflict of interest.

**Acknowledgements**

This work was supported by NASA Modeling, Analysis, and Prediction (MAP) Program (award
numbers: 80NSSC21K1496, NNH10ZDA001N, and 16-MAP16-0149). Computing resources for
the model runs were provided by the NASA High-End Computing (HEC) Program through the
NASA Center for Climate Simulation (NCCS) at Goddard Space Flight Center. We thank Dr.
Pierre Gentine of Department of Earth and Environmental Engineering, Columbia University, for
his help in GPP data and model validation, and Dr. Anastasia Romanou of NASA Goddard
Institute of Space Studies for discussions of ModelE structure.

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

**Tables**

**Table 1 Plant functional types used in BiomeE**

| Plant functional types | $V_{cmax}$ | LMA (kg C m$^{-2}$) | $L_{max,0}$ | $\rho_W$ (kg C m$^{-3}$) | $\alpha_Z$ | $T_{0,c}$ | $\beta_{0,D}$ | PS pathway |
|---|---|---|---|---|---|---|---|---|
| 1. Tropical evergreen broadleaf | 18 | 0.07 | 4.8 | 360 | 30 | 15 | 0 | C$_3$ |
| 2. Temperate/boreal evergreen needleleaf | 18 | 0.14 | 4.8 | 300 | 30 | -80 | 0 | C$_3$ |
| 3. Temperate/boreal deciduous broadleaf | 22 | 0.025 | 4.5 | 350 | 30 | 15 | 0 | C$_3$ |
| 4. Tropical drought deciduous broadleaf | 20 | 0.03 | 4.5 | 250 | 30 | 15 | 0.2 | C$_3$ |
| 5. Boreal deciduous needleleaf | 20 | 0.03 | 4.0 | 300 | 30 | 15 | 0.0 | C$_3$ |
| 6. Cold shrub | 18 | 0.025 | 3.0 | 360 | 20 | 15 | 0.1 | C$_3$ |
| 7. Arid shrub | 18 | 0.03 | 3.0 | 360 | 20 | 15 | 0.1 | C$_3$ |
| 8. C3 grass | 20 | 0.025 | 2.5 | 90* | 10 | 5 | 0.2 | C$_3$ |
| 9. C4 grass | 15 | 0.025 | 2.5 | 90* | 10 | 5 | 0.2 | C$_4$ |

$V_{cmax}$: leaf maximum carboxylation rate; LMA: leaf mass per unit area, $L_{max,0}$: is crown
maximum leaf area index; $\rho_W$: wood density; $\alpha_Z$: Height coefficient; $T_{0,c}$: Critical temperature for
phenology offset; $\beta_{0,D}$: critical soil moisture index for the offset of phenology; PS:
photosynthesis pathway
*Grass stem carbon density is calculated as tissue carbon divided by stem volume. The tissue
density of grass's stems can be as high as wood.




**Table 2 Sites for simulated ecosystem development illustration**

| Site | Dominant PFT | Coordination | Mean Temperature (°C) | Annual Precipitation (mm) |
|---|---|---|---|---|
| Bonanza Creek (BNC) | Broadleaf deciduous | 63.92°, -145.38° | -3.1 | 269 |
| Manitoba old black spruce site (MNT) | Evergreen needleleaf | 55.88°, -98.48° | -3.2 | 520 |
| Harvard Forest (HF) | Broadleaf deciduous | 42.54°, -72.17° | 8.5 | 1050 |
| Oak Ridge (OKR) | Broadleaf deciduous | 35.96°, -84.29° | 13.7 | 1372 |
| Konza LTER (KZ) | $C_4$ grass | 39.08°, -96.56° | 12.4 | 835 |
| Sevilleta LTER (SV) | Arid shrub | 34.36°, -106.88° | 12.7 | 365 |
| Walnut Gulch Kendall (WGK) | Arid shrub | 31.74°, -109.94° | 17.7 | 350 |
| Brazil Tapajos (TPJ) | Broadleaf evergreen | -2.86°, -54.96° | 26 | 1820 |


**Figure captions**

**Figure 1 Schematic diagram of the coupling of BiomeE into ModelE**
Panel A shows the structure of carbon and nitrogen pools and fluxes, and the interactions of BiomeE with TerraE, the land surface model in ModelE. The lines are the flows of carbon (green), nitrogen (brown), and coupled carbon and nitrogen (black). The green box is for carbon only. The brown boxes are nitrogen pools. The black boxes are for both carbon and nitrogen pools. Panel B shows the processes of plant physiology, demography, and crown organization in BiomeE.

**Figure 2. Prescribed global distribution of plant functional types.** Data are from the Ent global vegetation structure map.

**Figure 3. Vegetation structural dynamics with the full demographic BiomeE at the field sites listed in Table 2.** Critical height (panel b) is an index of the model PPA, which separates the trees that are in full sunlight if taller than critical height and those that are fully shaded if shorter than critical height.

**Figure 4. Site ecosystem development simulated by BiomeE with full demography for the field sites listed in Table 2.** GPP: gross primary production (kg C m$^{-2}$ year$^{-1}$); Plant C: vegetation biomass (kg C m$^{-2}$); Soil C: soil organic matter (kg C m$^{-2}$)

**Figure 5. Seasonal patterns of LAI and gross primary production (GPP) in the sample grids.** Two years of data are shown in this figure. The key to location abbreviations is in Table 2.

**Figure 6. Spatial patterns of LAI and gross primary production (GPP) in January and July simulated with the full demography model setting**. Panels a and b are the LAI and GPP of January in the year of 600 (the last year of model run). Panels c and d are July's in the same year.

**Figure 7. Spatial patterns of BiomeE (full demography) simulations and those from data**. "Obs." means different ways retrieved from observations. Obs. LAI is from Ent vegetation data

(Modis LAI 2004) (Ito et al., 2020; Tian et al., 2003). Obs. GPP is derived from Solar Induced
Fluorescence (SIF) data with a machine learning approach (Alemohammad et al., 2017). The
data are available from Jan. 2007 to Dec. 2015. The tree height data are from spaceborne light
detection and ranging (lidar) global map of canopy height at 1-km spatial resolution developed
by Simard et al. (2011). Biomass data are from Hengeveld et al. (2015). Soil carbon data are
from FAO Harmonized World Soil Database (version 1.2), updated by Wieder (2014).

**Figure 8**. **Grid comparison of full demographic BiomeE simulations with observations**
**estimates**. The red line in each panel is the 1:1 line. The data used in this figure are the same as
those in Figure 7.

**Figure 9 Site-level comparison with MsTMIP models**. The BiomeE predictions are from the
full demography. The abbreviations of the 8 sites (corresponding to model grid cells) and their
coordination, dominant PFTs, and climatic conditions are in Table 2. (See Figure S12 in SI for
the single cohort BiomeE simulations.)

**Figure 10 Latitudinal patterns of GPP, NPP, Biomass, and soil carbon as simulated by**
**BiomeE (with full demography) and MsTMIP models.** 'MIP Mean' is the mean of the six
MsTMIP model simulations. (See Figure S13 in SI for the single cohort BiomeE simulations.)

**Figure 11 Comparison between the simulations of the full demography and the single**
**cohort settings of BiomeE**. LAI: leaf area index; GPP: gross primary production ($kgC/m^2/year$);
Height: maximum tree height; Plant C: vegetation biomass ($kg\ C\ m^{-2}$); Soil C: soil organic
matter ($kg\ C\ m^{-2}$); Rh: heterotrophic respiration rate ($kg\ C\ m^{-2}\ year^{-1}$).

**Figure 12 Spatial patterns of the differences between the simulations of the BiomeE**. $\delta$
means the difference between the simulations of the full demography and the single cohort
models. LAI: leaf area index; GPP: gross primary production ($kg\ C\ m^{-2}\ year^{-1}$); Plant C:
vegetation biomass ($kg\ C\ m^{-2}$); Soil C: soil organic matter ($kg\ C\ m^{-2}$)

**Figure 13. Simulated competitively dominant PFTs at different total ecosystem nitrogen.**
The simulations were set as nitrogen-closed (i.e., no input and output of nitrogen). The number
in the title of each panel is the initial soil nitrogen. We used five PFTs that only differed in their
LMA ($\sigma$) and target root/leaf area ratio ($\varphi_{RL}$) corresponding to each LMA in each simulation.
Basal area (the sum of all trees' trunk cross sectional area) is used as the index of dominance.


**Figures**

**Figure 1**

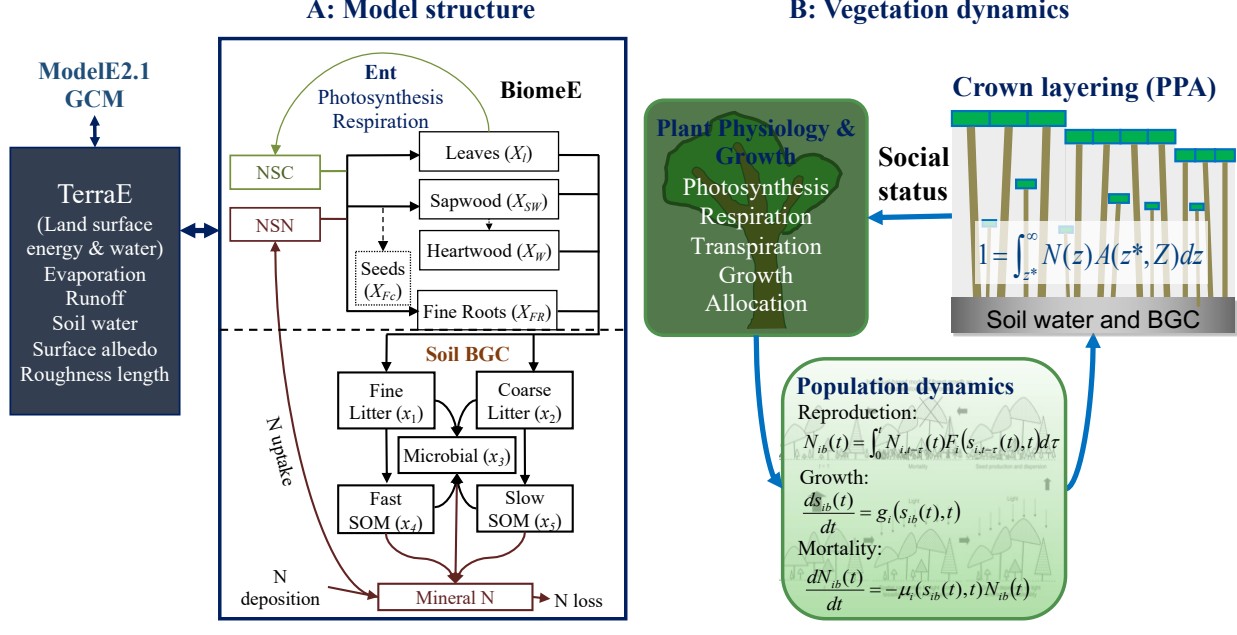


**Figure 2**

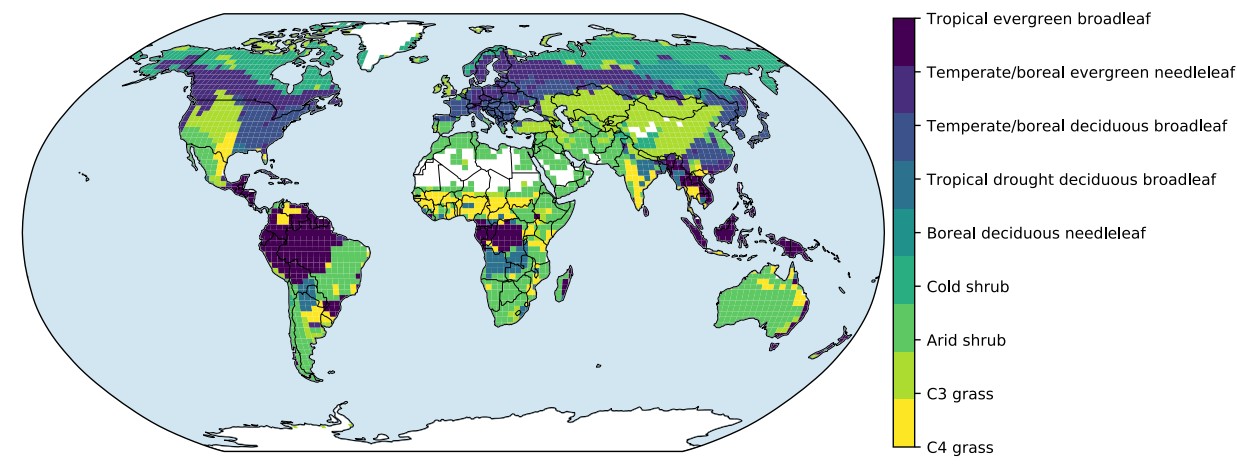


**Figure 3**

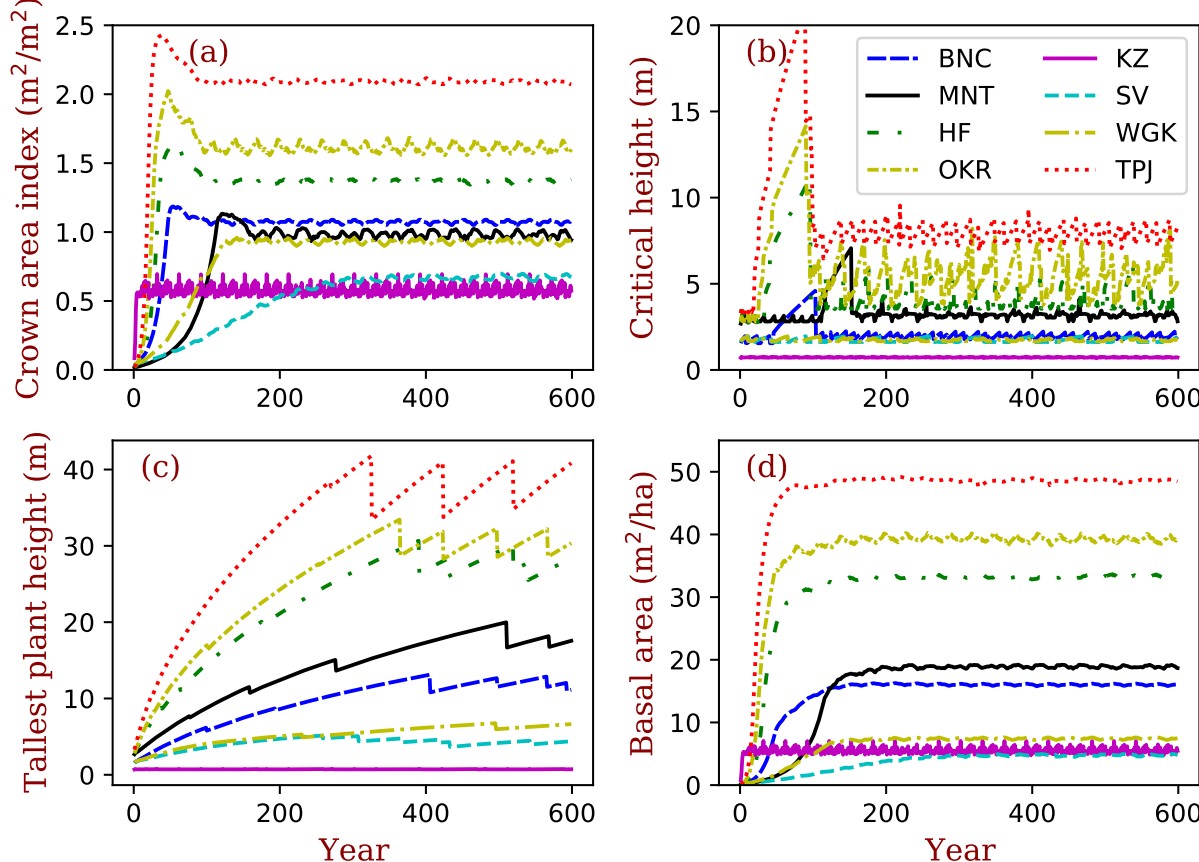


**Figure 4**

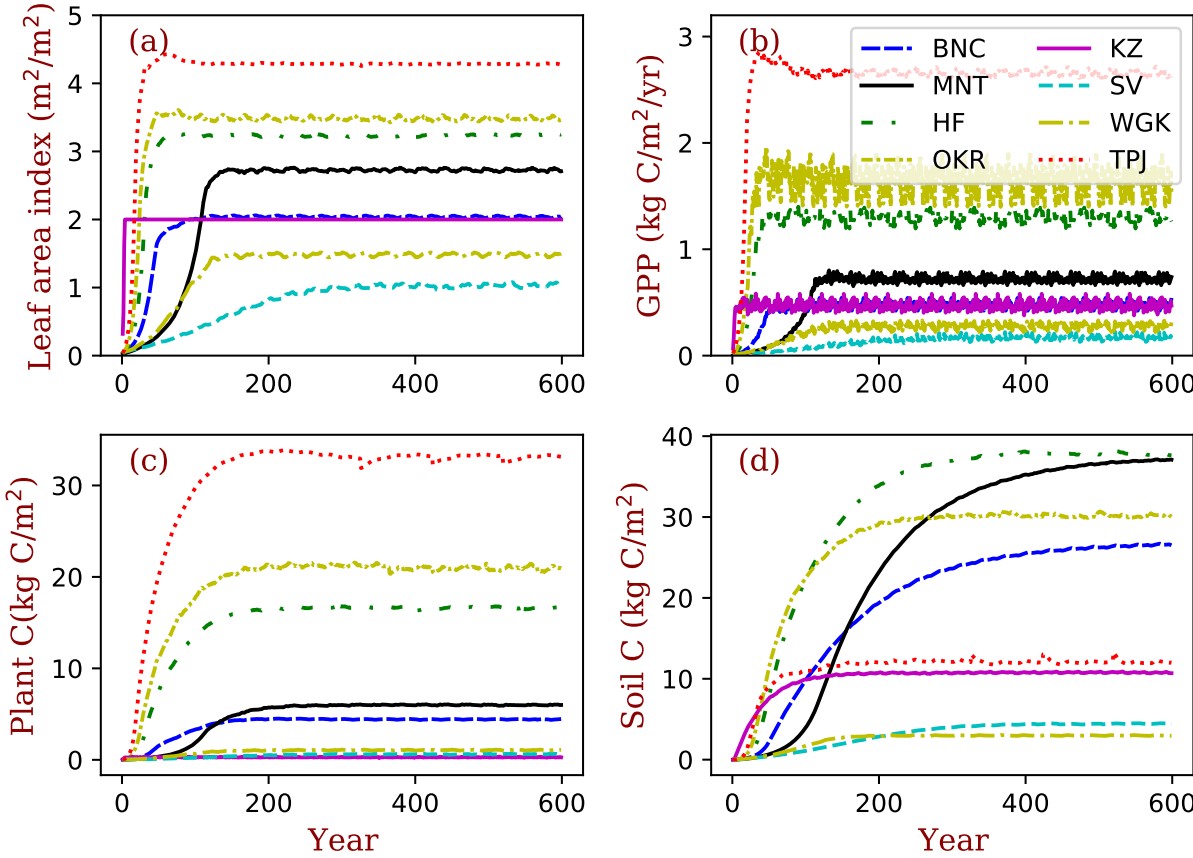


**Figure 5**

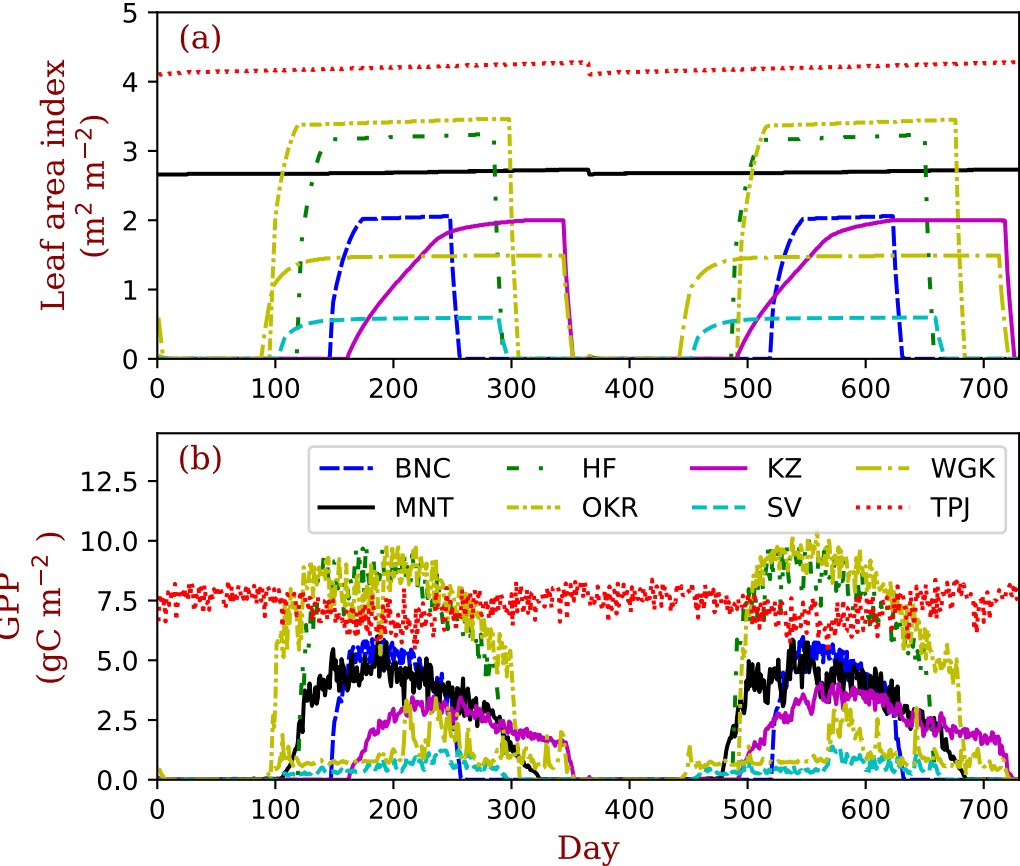


**Figure 6**

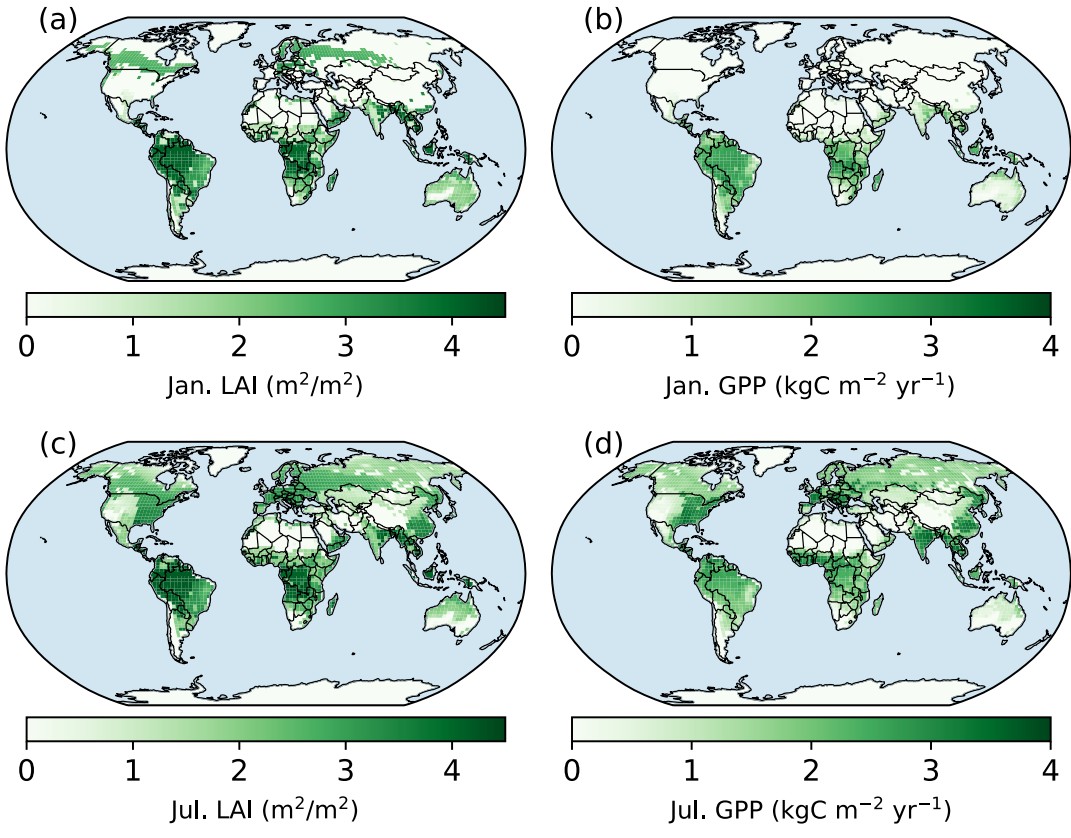


**Figure 7**

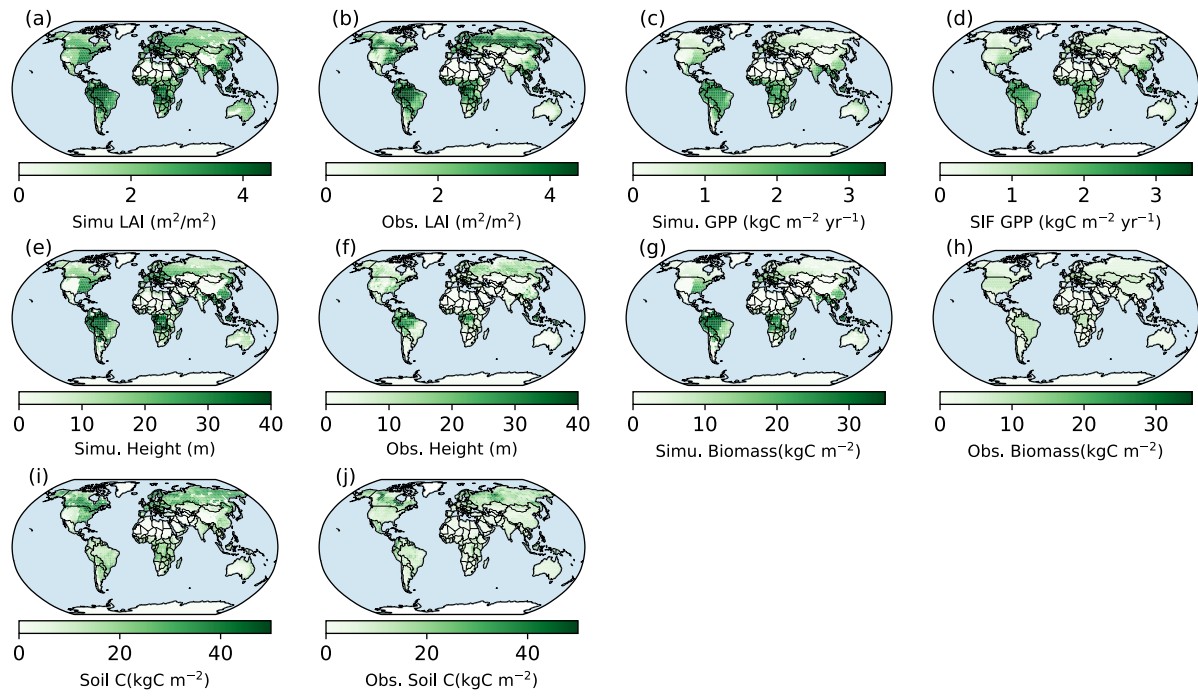

 **Figure 8**

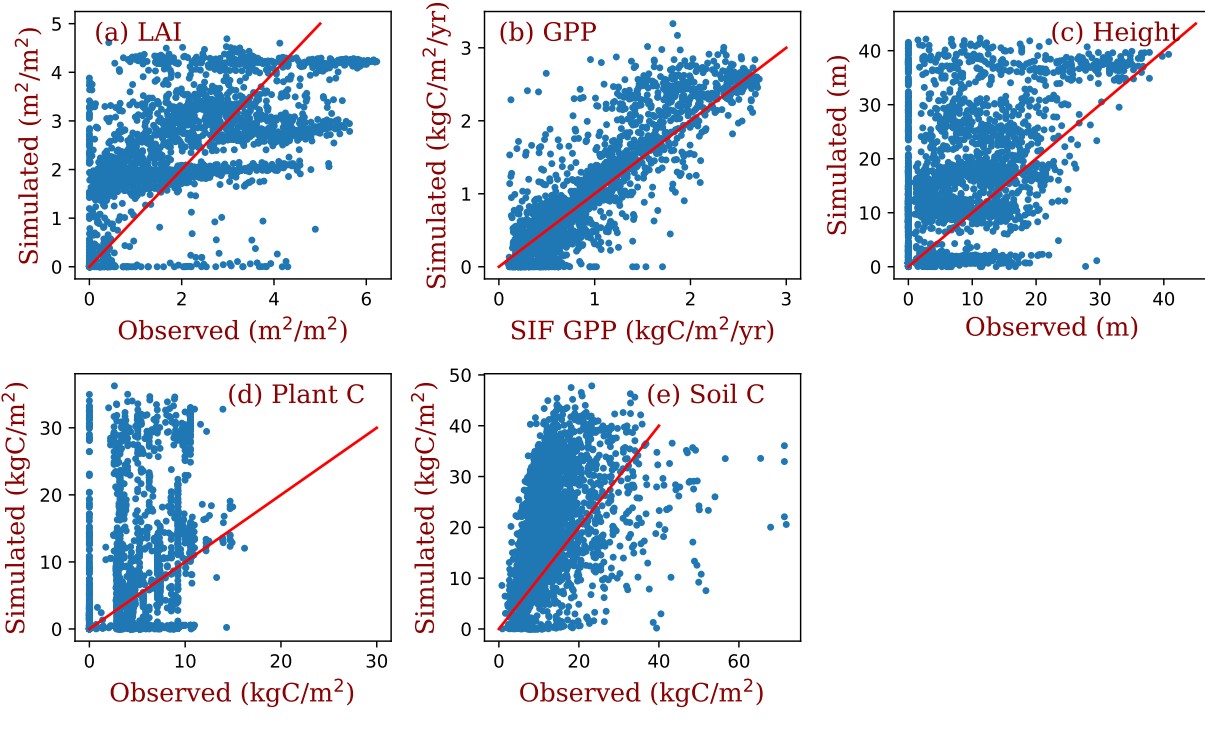

 **Figure 9**

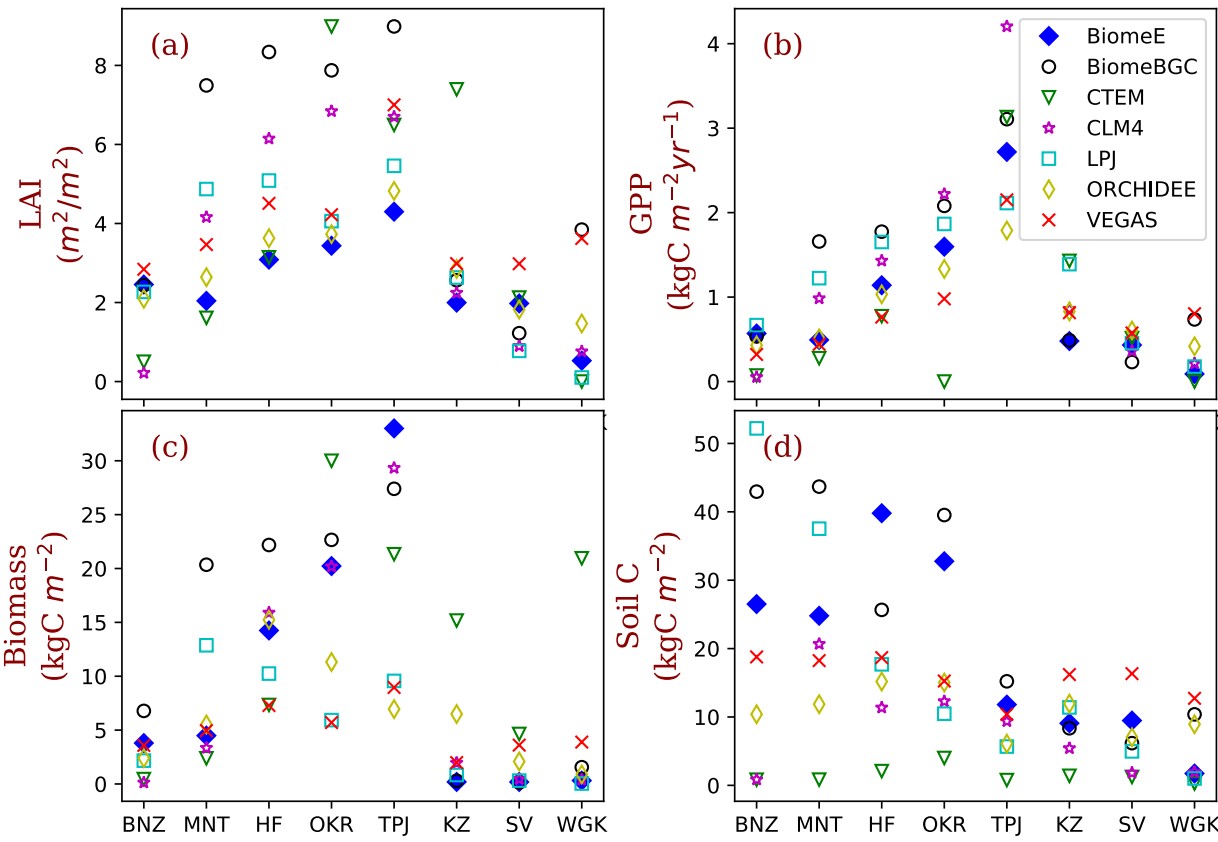

**Figure 10**

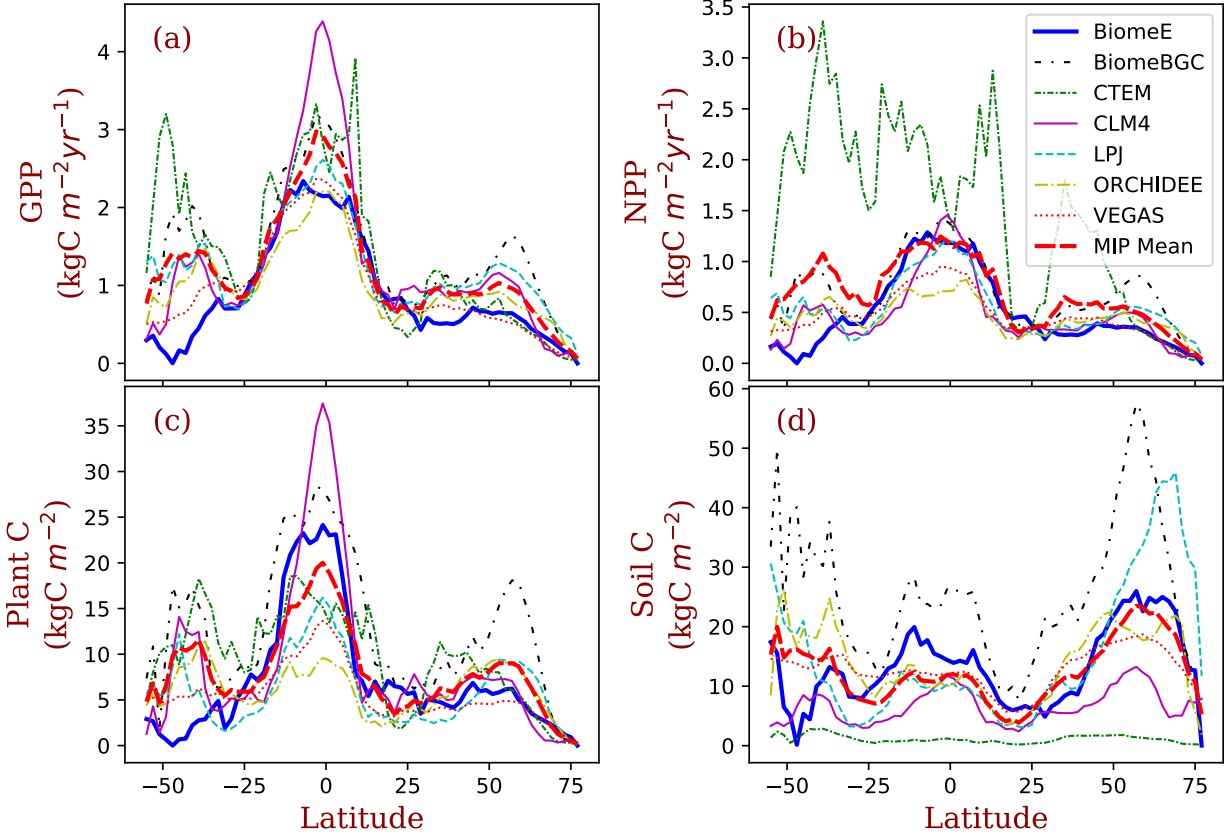


**Figure 11**

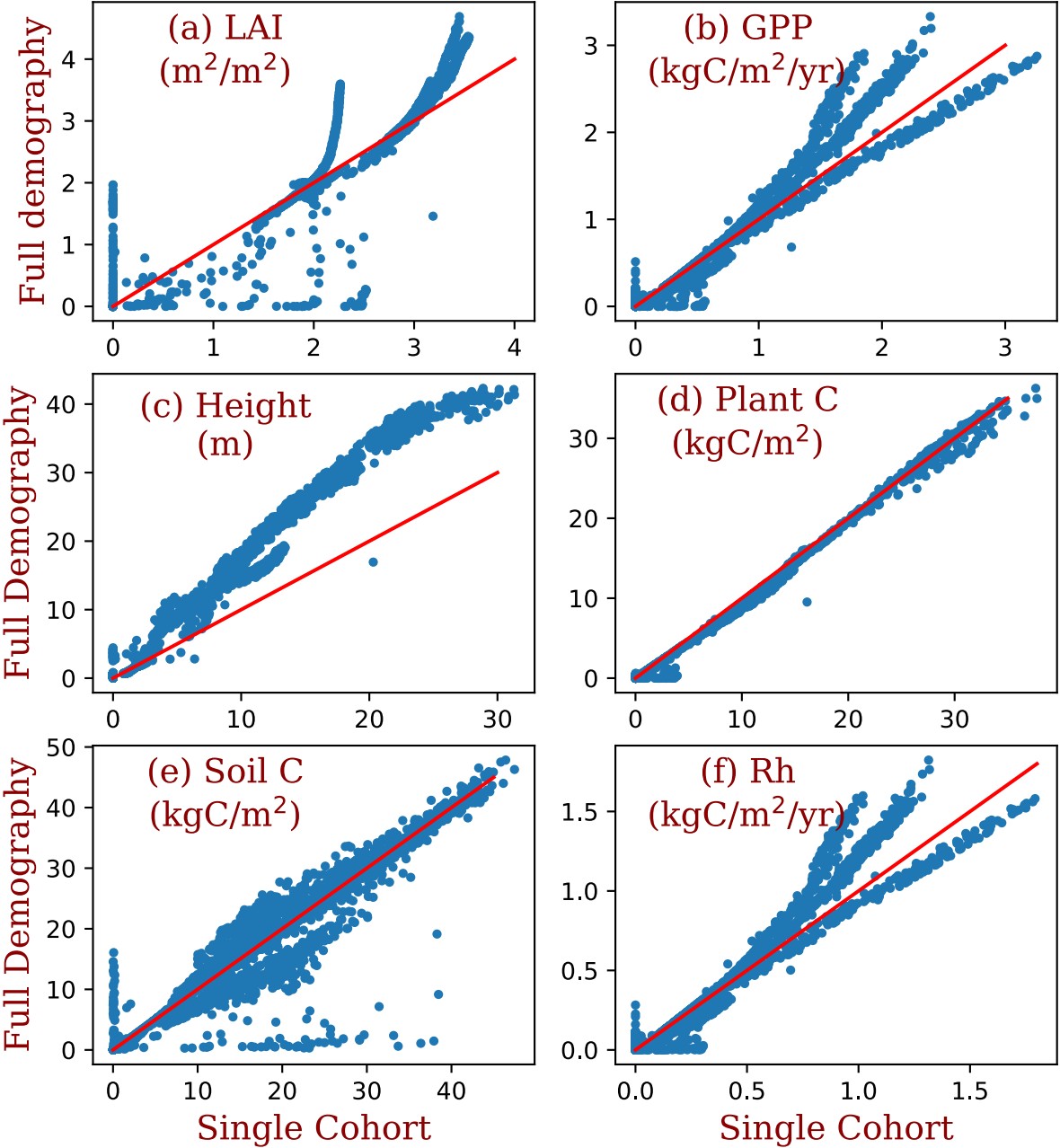


 **Figure 12**

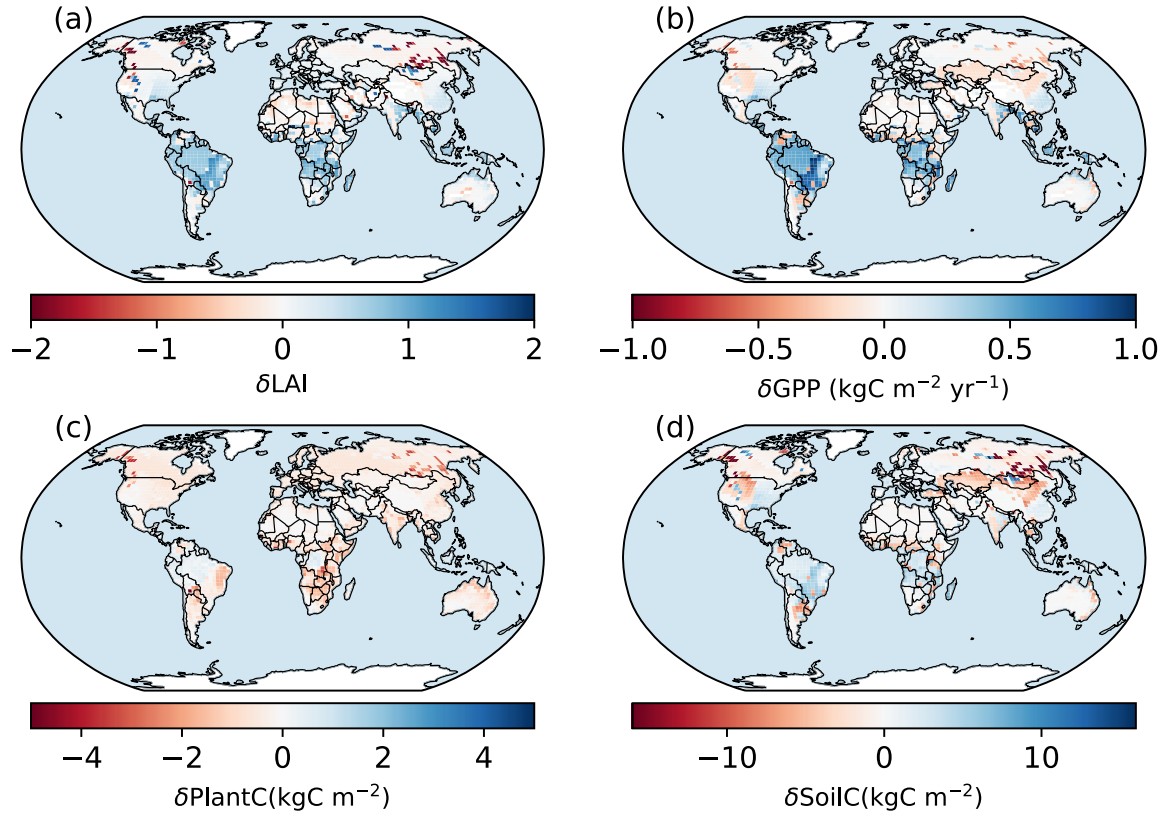


**Figure 13**

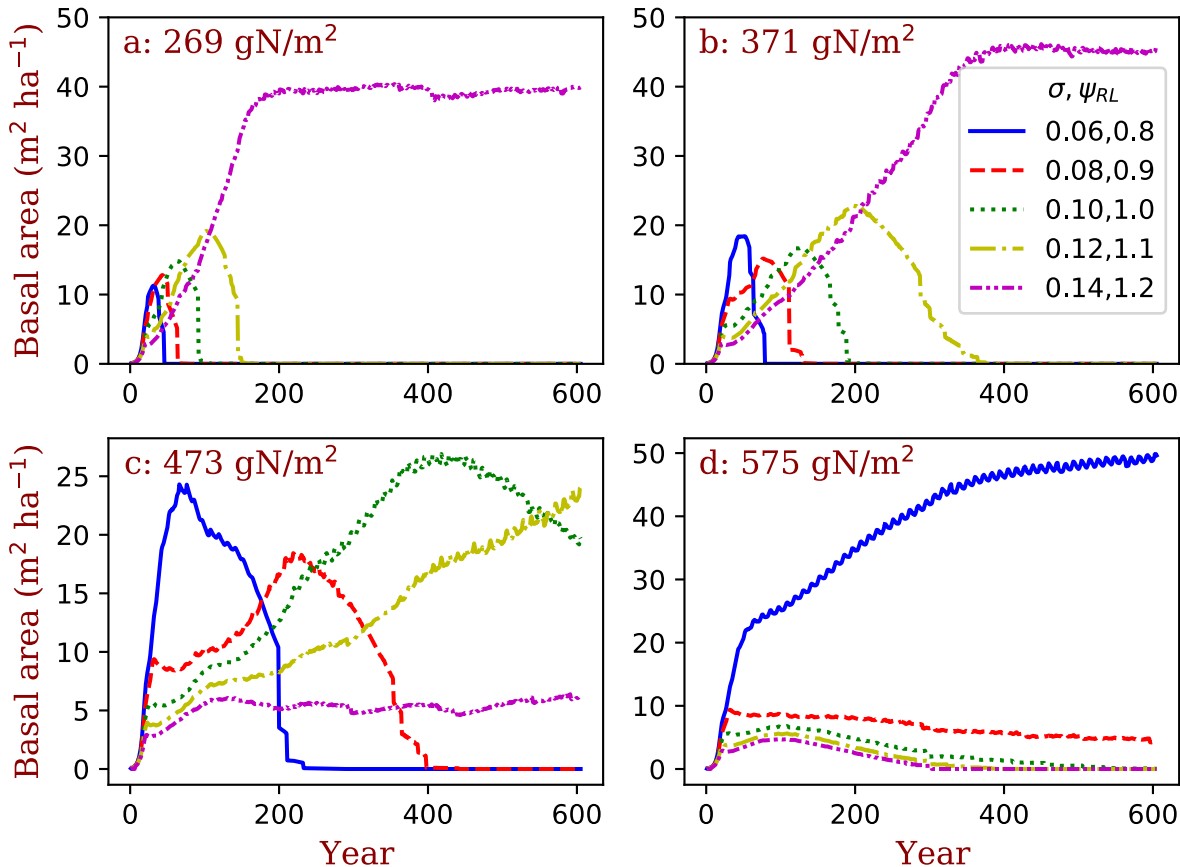
