# Peer review of "Ensheng Weng1,2, Igor Aleinov1,2, Ram Singh1,2, Michael J. Puma1,2, Sonali S. McDermid3,"

_Geoscientific Model Development, 2022_

## Author Response (AR1)

**Responses to Review comments (GMD- 2022-72)**

To the Editor:

Please find, below, our point-by-point response to the reviewer comments on our manuscript, "Modeling demographic-driven vegetation dynamics and ecosystem biogeochemical cycling in NASA GISS's Earth system model (ModelE-BiomeE v.1.0)". We have conducted extensive revisions in response to the reviewer's critiques, mainly by doing the following:

1) Moved the description of ModelE to the Method section.

2) Added details on the simulated vegetation structure and the results of the single cohort setting simulations into the Supplementary Information.

3) Added a sensitivity test to illustrate the model's capacity for predicting eco-evolutionary optimality of plant functional types (PFTs).

4) Added a section in the discussion about the model's performance in comparison to the MsTMIP models.

We have also addressed the other minor concerns brought up by the reviewers, and edited the text throughout the paper. We attached two files in resubmission, a revision-marked version and a clean version. Since the citation links were lost when we used the google doc to coordinate the revision, we only updated the citations in the clean version. There are some minor fixations in format and grammar that are not in the marked version.

Our responses to each comment are detailed below, with the reviewer comments in italics and our responses in plain text. We thank the reviewers for their thoughtful comments, and the editor for considering our manuscript.

Best wishes,

Ensheng Weng and Ben Cook on behalf of all authors

**Reviewer 1**

*This manuscript by Weng et al reported a model improvement. I am happy to read it and know many great improvements. However, there are some unclear things that need improve before publication.*

*Line 47-49: this sentence is not necessary.*

Removed.

*Line 86-99: should this paragraph be moved to model introduction section?*

We moved it to the model description section.

*Line 213-214: how to determine soil moisture threshold is quite important. However, it is difficult to understand how to determine this threshold. the authors need introduce more about this.*

In this paper, we only slightly tuned this parameter to make the drought-deciduous PFTs to match the drought phenology induced by seasonal precipitation patterns. A plant hydraulic module is currently being developed for this model to more explicitly link soil water and plant physiology. We added a brief explanation of how we tune the critical drought index in the revised paper.

"The critical soil moisture values that trigger new leaf growth and leaf fall are defined as PFT-specific parameters. We slightly tuned these two parameters according to the soil moistures where the deciduous PFTs' leaves start to grow or fall. Usually, the critical soil moisture for starting new leaf growth is higher than the soil moisture levels that trigger leaf fall so that the plants can have a stable growing season."

*Line 219: what is structural biomass? What are the parameters of αc, αz, θc and θz.*

The "structural biomass" is referred to as sapwood plus heartwood. We replaced it with "woody biomass (sapwood plus heartwood)" to make it easier to understand.

*For Eq. (5), the most important thing is how to simulate D?*

Yes. dD/dt (and its integral D(t)) is the process that links carbon fluxes to vegetation structure. It also bridges the traditional biogeochemical cycle model, which assumes an ecosystem can be described by fluxes and pools, to the demographic models that explicitly simulate the three-dimensional growth of trees (tree diameter, height, and crown area). We have a paragraph to explain it after Eq (7), and clarified the meaning of predicting D in the revised manuscript.

"This equation transforms the carbon gain from photosynthesis to the diameter growth that results from wood allocation and allometry (Eq 5). With an updated tree diameter, we can calculate the new tree height and crown area using allometry equations, and the targets of leaf and fine root biomass (Eq. 5)."

*Line 231: is that correct to keep a minimum growth rate of stems? Why do the authors set the equation like this?*

And

*Line 232-235: it is difficult to understand how the authors simulate carbon allocation. Carbon allocation is quite hard to simulate indeed, and especially for demo-type model, the authors should pay more attention to impacts stand age on carbon allocation. Please refer the Xia et al. 2019.*

Yes. The purpose is to keep the consistency of the vascular system in leaves, sapwood and roots, though they are separated into three functional organs. We discussed this assumption in the manuscripts describing the previous versions of BiomeE and LM3-PPA (Weng et al. 2015 and Weng et al. 2019). We edited this paragraph and added new explanations in the revised manuscript to explain why we designed such an allocation scheme.:

"**Plant growth and allocation of carbon and nitrogen to plant tissues**

The allocation of NPP wood, leaves, and roots is affected by climate and forest age (Xia et al., 2019; Litton et al., 2007). However, vegetation models cannot capture these patterns well at large spatial scales, even if the adaptive responses to climate and forest ages are considered (Xia et al., 2019, 2017), partly because of the absence of explicit representation of shifts in species composition and competition between individuals (Dybzinski et al., 2015; Franklin et al., 2012). BiomeE has an optimal growth scheme that drives the allocation of carbon and nitrogen to leaves, fine roots, and stems based on the optimal use of resources and light competition (Weng et al., 2019). In this scheme, the growth of new leaves and fine roots follows the growth of woody biomass (i.e., stems), and the area ratio of fine roots to leaves is kept constant during the growing season. The allocation of available carbon between structural (e.g., stems) and functional (e.g., leaves and fine roots) tissues is optimal for light competition at given nitrogen availability.

Mathematically, differentiating the stem biomass allometry in Eq. 5 with respect to time, using the fact that $dS/dt$ equals the carbon allocated for wood growth ($G_W$), gives the diameter growth equation:

$$\frac{dD}{dt} = \frac{G_W}{0.25\pi\Lambda\rho_W\alpha_Z(2+\theta_z)D^{1+\theta_Z}} \qquad (7)$$

This equation transforms the carbon gain from photosynthesis to the diameter growth that results from wood allocation and allometry (Eq 5). With an updated tree diameter, we can calculate the new tree height and crown area using allometry equations, and the targets of leaf and fine root biomass (Eq. 5). Generally, the growing-season average allocations of carbon and nitrogen to different tissues are governed by two parameters: the maximum leaf area per unit crown area ($l_{max}$) and fine root area per unit leaf area ($\varphi_{RL}$) (Eq. 5). The optimal-growth allocation scheme combined with explicit competition for light and soil resources in our model makes it possible to simulate the underlying processes that determine emergent allocation patterns (Weng et al., 2019; Farrior et al., 2013; Dybzinski et al., 2011; Farrior, 2019).

"

*Line 246 "Reproduction and Mortality". They are very important and hard to simulate. I am happy to see the authors made great contributions.*

Thanks!

*Line 258: cannot understand "U-shape" curve?*

"U-shape" means the highest mortality rates occur for seedlings and old trees. We reworded this sentence with the definition of "U-Shape".

"These factors result in high mortality rates of seedlings and old trees (i.e., a "U-shaped" mortality curve)"

*Eq. (9) it will be better to introduce the basic principle.*

This equation delineates the mortality rate that changes with social status (crown layers), shade effects, and tree sizes. We added more explanations in the revised manuscript.

"These factors result in high mortality rates of seedlings and old trees (i.e., a "U-shaped" mortality curve). We use the following equation to delineate a mortality rate that varies with social status (crown layers), shade effects, and tree sizes:

$$\mu(s,t) = \mu_0(1 + f_L f_s)f_D \tag{9}$$

where $f_L$ is the shade effects on mortality ($f_L = \sqrt{L-1}$), $f_S$ is seedling mortality when a tree is small ($f_s = A_{SD}e^{-B_{SD} \cdot D}$), and $f_D$ represents the size effects on the mortality of adult trees ($f_D = m_s \frac{e^{A_D(D-D_0)}}{1+e^{A_D(D-D_0)}}$)."

*Figure 2. it will be helpful to give name of each vegetation type in the figure caption. Did you include cropland?*

Done as suggested. And we moved the question to Fig. 7 (about cropland) here

*Figure 7. how did the authors treat cropland? If the model scheme impacts this global comparison?*

Cropland is not included in this paper because our purpose is to describe a model for natural vegetation. We only made test runs with potential vegetation as represented by the 9 PFTs.

It does affect global comparisons of biomass. Please see the answer to the question about Figure 8).

*Line 355: what do you mean "The interpolation of radiation"?*

The forcing data is at six-hour time step. We interpolate them into hourly/half-hourly time step. We clarified it in the revised manuscript:

"We interpolated the radiation data ($R_S$) into half-hour timesteps based on the sun zenith angle ($\theta_S$) and radiation penetration rate calculated from data."

*Figure 4. can you explain why there are sharp decreases of simulated height? I am also confused why the crown area index increased first and then decrease?*

The sharp decrease in critical height indicates the transformation from even aged trees to mixed-age trees in the canopy. "Critical height" is the shortest tree in the canopy layer.

We added a paragraph to explain the structural dynamics of vegetation in the simulations and switched Figures 3 and 4 to make it easier to understand.

"In the forest sites, the simulated vegetation structure by the full demographic model changes with the growth, regeneration, and mortality processes (Fig. 3). It can be separated into three stages according to the canopy crown dynamics: 1) open forest stage, 2) self-thinning stage, and 3) stabilizing stage.  In the open forest stage, the crown area index (CAI) is less than 1.0 and all the individuals are in full sunlight. The tree crowns grow rapidly to occupy the open space (Fig. 3: a). In the self-thinning stage, the open space is filled by the crowns of similar sized trees (i.e., the forest is closed) and canopy trees are continuously pushed to the lower layer(s) (i.e., self-thinning) and the CAI continues to increase due to the limited space with growing tree crowns (i.e., the new spaces vacated from the canopy trees' mortality cannot meet the space demand from crown growth).  The sizes of trees in the canopy layer are still similar in this period (Fig. 3: b and c) and the critical height (the height of the shortest tree in the canopy layer) keeps increasing in this period. In the stabilizing stage, when the space generated by the mortality of canopy trees is larger than the growth of canopy tree crown area, no trees are pushed to the lower layer and the lower layer trees start to enter the canopy layer and fill the space, leading to a sharp

decrease in critical height (Fig. 3: b) and the mixing of different sized trees in the canopy layer. The CAI is decreasing as well because of the high mortality rates of the understory layer trees. As time goes on, the growth, regeneration, mortality, and space filling processes are equilibrated, and the forest structure is then stabilized."

Additionally, to improve the clarity of simulated model structure, we plotted the cohort structure for all the 8 sample sites in the Supplementary Information (SI).

*Figure 5. simulated LAI is not good enough, but I totally understand it is very hard task. You may discuss this issue, and especially point out how we should improve LAI simulations in the further studies.*

Done as suggested. We have two paragraphs discussing LAI issues in the revised manuscript (**5.2 Model predictions and performance**)

LAI is an illustrative variable for understanding why compromises are necessary when integrating ecologically based vegetation models into ESMs. LAI, as a critical prognostic variable in vegetation models, links both plant physiology and biogeophysical interactions with climate systems. While LAI is usually simulated by a fixed allocation scheme, even if the allocation ratios are dynamic with vegetation productivity (Montané et al., 2017), the prediction of LAI in models is often simplified as the balance between growth and turnover. Modelers tend to tune LAI to fit observations and get the required albedo and water fluxes whatever their parameters of photosynthesis and respirations are. This LAI usually makes the lower layer leaves carbon negative. However, a first principle is that a tree should have an optimal LAI to maximize its carbon gain as a result of crown structure, light interception, and community-level competition (Anten, 2002; Hikosaka and Anten, 2012; Niinemets and Anten, 2009). Thus, in our model, because of the assumption of the uniform leaves within a crown, we defined a much small target LAI to avoid carbon negative leaves.

The "uniform leaf" assumption makes the lower layer leaves carbon negative when LAI is tuned close to that observed in tropical and boreal evergreen forests (where LAI is around 5~7). Therefore, the photosynthesis rate must be tuned to fit the canopy photosynthesis by keeping the carbon negative leaves. However, the carbon negative leaves do not affect ecosystem dynamics

in the "single-cohort" models because the whole canopy net carbon gain is still reasonable and can be fitted to the observed dynamics. This contrasts with the demographic version of the model, which represents trees with different sizes and in different layers and creates conditions where seedlings in the understory cannot survive because of light limitation and negative carbon balances in some dry and cold regions. The leaf traits in the crown profile should, in reality, be a function of light, water and nitrogen (Niinemets et al., 2015). A more complex crown development module will then be required to simulate branching and leaf development and deployment processes. Modelers should balance the model complexity and computing efficiency then.

*Figure 8. these results are surprised for me. I thought the model can simulate plant carbon better than soil carbon. But it seems that I am not correct. Would you like please to explain the reason for large uncertainties of plant carbon simulations?*

We simulated the potential vegetation biomass without considering land use changes and disturbances. However, the biomass data we used for validation includes the effects of land use and disturbance, which are absent in our simulations. We clarified it in the revised manuscript and explained why our simulated carbon agrees with data better than the simulated biomass does.

"Simulated biomass is much higher than the observations because, in the observations, many forest regions have been transformed to low biomass land use types (such as croplands) or represent earlier successional stages with less accumulated carbon (i.e., not equilibrium states). Simulated soil carbon does track the observations better than biomass, likely because soil carbon stocks are more stable compared to biomass; and GPP does not change much compared to the changes in vegetation biomass because leaves can reach to equilibrium much faster than the biomass does (Fu et al., 2017). For areas where the model underpredicts soil carbon, the difference could arise from the missing biogeochemical processes that may lead to high carbon accumulation in some regions (e.g., peats) (Davidson and Janssens, 2006; Briones et al., 2014; Euskirchen et al., 2014) or the relatively high uncertainties in the soil carbon data (Tifafi et al., 2018)."

**Reviewer 2**

*The authors present a model development work on vegetation demographics and seek to implement it into an Earth system model. The new model features include a greater diversity of global plant functional types and a new phenological scheme. They also compare the behaviour of this model to the eight observational locations and six MsTMIP simulations. As seen from the results within the paper it is possible to capture vegetation structure and dynamics reasonably even within such a parsimonious model at a global scale. Overall, this work is well structured and is useful in highlighting some new issues in improving the representation of the terrestrial carbon cycle within ESMs. I think the manuscript could be publishable after some major revisions. I have a few comments below.*

Thanks!

**Major comments:**

*1. One of the most representative feature of this model is the full demographic processes. The authors mainly compare the differences between the simulations of the full demography and the single cohort settings of BiomeE. I think authors should separately compare simulations of these two versions of BiomeE with observations, if possible, to show the advantages of full demography in reproducing ecosystem dynamics (Figs. 6-10).*

We have added figures comparing the single cohort model simulations with observations. Since these results are similar, at the global-scale, to the full demography simulations, these results are in the supplementary information. We used one figure (Fig. 11) in the revised manuscript to show the differences between the full demography and single cohort settings, and discussed the advantages of the representation of full demography in a newly added section "5.4 Insights from comparison with MsTMIP model", which also is a response to reviewer 2's major comments 6 and 9.

The new sensitivity analysis in response to major comment 3 also shows some of the advantages of the full demography model.

*2. Lines 157-158: "A set of continuous plant traits are used to define the distinctive plant types".*
*Please specify the continuous trait assignments for plant functional types, especially the*
*differences with traditional PFT-based model.*

The trait assignments for plant functional types are in Table 1. We reworded this paragraph to make it easier to understand (copied blow):

"In this model, we use a set of continuous plant traits to define plant functional types, so that we can simulate plant emergent properties (such as dominant plant types, vegetation compositional changes, etc.) in response to climate changes based on the underlying plant physiological properties and ecological principles through eco-evolutionary modeling in the future. For example, life forms are defined by the continuums characterized by wood density (woody vs. herbaceous), height growth coefficient (tree vs. shrub), and leaf mass per unit area (LMA, for evergreen vs. deciduous). Deciduousness is defined by cold resistance (evergreen vs. cold deciduous), and drought resistance (evergreen vs. drought deciduous). Grasses are simulated as tree seedlings with all stems senescent along with leaves at the end of a growing season. The individuals are reset back to initial size each year and the population density is also reset using the total biomass of current cohort and predefined initial size of grasses. The photosynthesis pathway is predefined as C3 or C4."

The newly added sensitivity test also serves as a case study illustrating this concept (see response to Comment 3 below) and its advantages over the multiple species summary-based definition of PFTs in eco-evolutionary modeling.

*3. To represent the major variations in plant functional diversity, the authors chose four plant traits as the primary axes to define PFTs: wood density, leaf mass per unit area (LMA), height growth parameter, and leaf maximum carboxylation rate (Vcmax). I would suggest **a sensitivity analysis of these plant traits to different ecosystem functions**, which would be very instructive for further model improvement and localization.*

We conducted a sensitivity test at one site (Oak Ridge) to show model predictions of competitively optimal strategy at different ecosystem nitrogen levels. This is not a traditional

"sensitivity test" for those pool-flux models. Instead, it is a test of the sensitivity of the simulated competitively dominant plant types to changes in environments. We sampled 5 PFTs from two continuums of plant traits, leaf mass per unit area and root/leaf area ratio, and simulated their competition during succession and therefore the competitively dominant types (i.e., evolutionarily stable strategy, ESS) at four different environments represented by ecosystem total nitrogen.

This test shows the major advantages of this model in PFT definitions, modeling of vegetation compositional responses, and eco-evolutionary optimality. In the original manuscript, we discussed these properties of this model, without a case to show them. With this case simulation, those discussions would be more informative.

Please see section "**4.4 Eco-evolutionary simulation and sensitivity test**" for detail (copied below).

"This model has the potential to predict competitively dominant PFTs in the continuum of plant traits through succession simulations according to the principles of evolutionarily optimal competition strategy. We illustrate this with a set of simulations conducted at a series of ecosystem nitrogen content (from 269 to 575 g N/m$^2$) with five PFTs sampled from the continuums of LMA ($\sigma$, from 0.06 to 0.14) and target root/leaf area ratio ($\varphi_{RL}$, from 0.8 to 1.2 corresponding to each LMA). The different ecosystem total nitrogen represents the environmental conditions that can result from soil and climate conditions. The simulations are set as nitrogen-closed (i.e., no input and output of nitrogen). At the lowest ecosystem total nitrogen (Fig. 13: a), the PFT with highest LMA (0.14 kg C/m$^2$ leaf) wins. With increases in ecosystem nitrogen (Fig. 13: b~d), the winner shifts to lower LMA PFTs. This means that in infertile soils or cold climates with slower biogeochemical cycles (e.g., tundra and boreal forests), the eco-evolutionarily optimal PFTs should have high LMA leaves, and vice versa. This pattern is consistent with the predictions of a theoretical model derived in Weng et al. (2017). This simulation is also a case of sensitivity of simulated vegetation dynamics to environmental conditions. Vegetation can shift their compositions and dominant plant traits to maintain an eco-evolutionarily optimal state, and thus amplify or attenuate the responses of ecosystem carbon cycle to climate changes.

[Figure]

**Figure 13. Simulated competitively dominant PFTs at different total ecosystem nitrogen.** The simulations are set as nitrogen-closed (i.e., no input and output of nitrogen). The number in the title of each panel is the initial soil nitrogen. We used five PFTs that only differed in their LMA ($\sigma$) and target root/leaf area ratio ($\varphi_{RL}$) corresponding to each LMA in each simulation. "

*4. Methods Line 184: it is unclear to me about the assumptions in the phenological scheme. Why to define the nine PFTs as these four phenological types? Regarding the comparative advantage and competitiveness of deciduous vs. evergreen trees, are there any basic theories that evergreen species are more resistant to cold and drought than deciduous tree species? According to the global vegetation distribution, evergreen broadleaf species are usually distributed in warm and moist environments. What kinds of functional traits suggest that evergreen species are adaptive under water limitation and cold conditions?*

We only defined the possible factorial combinations of drought and cold deciduous, but did not discuss who will be more competitive. It is possible that the evergreen would be more competitive in high seasonality regions (e.g., evergreen in boreal regions), though the first response of plants to harsh environments (e.g., cold or dry) is often to shed their leaves. Our simplified definition of phenology is designed to make it possible to evaluate the competitively optimal strategy in future studies. We have added a more detailed explanation in this section.

"These phenological types represent different strategies of dealing with environmental stresses and pressure of competition. It is possible that the evergreen would be more competitive in high seasonality regions (e.g., evergreen in boreal regions), though the first response of plants to harsh environments (e.g., cold or dry) is to shed their leaves. Our definition of phenology is designed to make it possible to evaluate the competitively optimal strategy in future studies."

*5. Methods Line 254: PFT-specific parameterisations for the mortality parameter are used, so are there different PFTs each with their own cohort structure? How are PFT-specific background mortality parameters set in the model? Are they all come from observations across different vegetation types? Related reference is missing in the main text. Since the most size-dependent mortality research focus on closed-canopy forest system, whether the "U-shaped mortality pattern" can be extended to other vegetation systems, including forbs, shrubs, grasslands, systems with open canopies and systems experiencing different risks in different environment?*

The mortality pattern is delineated by the Eq. 9, as a function of size (diameter), social status (layers). We now explain this equation in detail in the revised manuscript (copied in the response to Reviewer 1's question to Eq. 9). The default mortality rates for different PFTs are fixed according to the general mortality patterns of trees in the forests across the world. We did not extensively tune these parameters.

Grasses represent a different survival strategy compared with trees, who must keep their trunk standing decades to hundreds of years. For grasses, we have a special definition for their behavior, including all leaves occupying one layer and a high turnover rate for stems. Consequently, grasses will not grow big enough to reach the point where large size-induced mortality happens.

*6. Methods Line 374: how does disturbance history set in the MsTMIP simulations? I'm wondering whether the large inter-model discrepancy in simulating plant biomass is caused by disturbance dynamics? For clarity, can the authors be a bit more explicit about the experimental design of the MsTMIP.*

According to Huntzinger et al. 2013, MsTMIP only provided prescribed land use types. It is up to the participant models for the disturbances. We clarified this in the revised manuscript.

"MsTMIP provided prescribed land use types for all the participant models. However, it is up to the participant models for disturbance impacts on ecosystems (Huntzinger et al., 2013). MsTMIP conducted five sets of experimental runs with different climate forcing, land-use history, atmospheric CO2 concentration, and nitrogen deposition. In this study, we used the SG1 simulation experiment because it is driven by the 1901~2010 climate forcing data with constant $CO_2$ concentration and constant land cover (Huntzinger et al., 2013), which are the closest to our model runs.

*7. Figure 4d: the authors point out that model analyses are based on equilibrium simulations without explicit disturbances. But the critical height across forests shows an abrupt decrease in the 100 years of model run. What reasons made this pattern happen in the model? Is that driven by the aging-related mortality of canopy trees? Could you discuss more the underlying mechanisms behind the emergent ecological phenomena?*

The "abrupt decrease" in the critical height is a behavior of the model. The simulated forest is transforming from even aged to mixed aged during this period. In the canopy layer, the trees are gradually replaced by younger trees as the old trees yield their space due to mortality.

We explained the details of forest succession in section "**4.1 Simulated vegetation structure and ecosystem carbon dynamics in different climate zones**" in the revised manuscript. The text has been copied in answering Reviewer 1's question to Figure 4.

We also included the detailed cohort data as figures in the SI (Figures S1~S8).

*8. Result Line 515-517: the formulation of allometry makes the tree height growth as a function of tree diameter (Eq. 5 in the main text). Since the two model versions have similar stem growth and tree size distribution, I would assume that tree height growth is stable as well. Why the full demography model shows higher tree height than the single-cohort model (Figure 11c)?*

For the full demography model, there are many cohorts with different heights and the vegetation height is based on the tallest tree (cohort). Most trees are much shorter than the tallest. For the single cohort model, all the trees have the same height because all trees are in one single cohort. So the tallest is shorter that it in the full demography model

We clarified this in the revised manuscript. The detailed cohort data (as figures) in the SI (Figures S1~S8) also show this.

*9. Result: the authors evaluate the model outputs with the MsTMIP simulations in the Result section. The simple intercomparison would be invaluable to help determine which model behaviour is more realistic. I think it would be interesting to have a section in the discussion tracing the variability that emerges among the models and informing what modeling structural choices or assumptions lead to improved model estimates. Since this paper is a model description paper, further discussion by model developers on the potential reasons for the biases would be much appreciated.*

We added a section in the discussions about why these models perform differently, especially focusing on the assumptions of our model, BiomeE.

" **5.4 Insights from comparison with MsTMIP model**

Most of the MsTMIP participant models have been analyzed by a model traceability method developed by Xia et al. (2013), which hierarchically decomposes model behavior into some fundamental processes of ecosystem carbon dynamics, such as GPP, carbon use efficiency (CUE), allocation coefficients, carbon residence time, carbon storage capacity, and environmental response functions (Zhou et al., 2021; Xia et al., 2013; Luo and Weng, 2011). This method is based on the assumptions of the linear system and the ecosystem emergent behavior per se (Emanuel and Killough, 1984; Eriksson, 1971; Sierra et al., 2018; Luo et al.,

2012), making it is consistent with the concepts that are used as the basis of ecosystem carbon cycle models. The analyses of model traceability found, for the carbon cycle dynamics, the major uncertainty is from the modeling of the turnover rates (reciprocals of residence time) of vegetation and soil carbon pools (Jiang et al., 2017; Chen et al., 2015). From CMIP5 to CMIP6, the modeling of NPP has been greatly improved, while the ecosystem carbon residence time remains highly biased (Wei et al., 2022).

According to the concepts of this traceability analysis approach (Xia et al., 2013), BiomeE also has a high uncertainty in the modeling of residence times of vegetation and soil carbon pools, because the mortality is picked up from the global forest data and the SOC decomposition processes are highly simplified. These issues have been discussed in the section of "5.3 Major uncertainties in BiomeE". These concepts (e.g., residence time, allocation coefficients) describe model emergent properties resulting from the underlying biological and ecological processes (i.e., micro-dynamics vs. macro-states).  Fitting the emergent properties directly to improve model behavior is natural and convenient because many vegetation models are using these emergent properties (e.g., CUE, residence time, and allocation coefficients) to describe ecosystem processes in their formulations as a tradition of ecosystem modeling.

There are a couple of common and long-lasting issues in terrestrial ecosystem modeling, such as responses to warming, responses to atmospheric $CO_2$, drought stress effects, and vegetation compositional changes (Harrison et al., 2021; Franklin et al., 2020; Luo, 2007). These issues represent our knowledge gaps in ecosystem ecology. For modeling vegetation dynamics eco-evolutionarily, we need to use the fundamental ecological processes and unbreakable physical rules to simulate the emergent processes (e.g., Weng et al., 2019; Scheiter et al., 2013), With the design of vegetation modeling in the BiomeE, such as the explicit demographic processes, individual-based competition for different resources, and flexible trait combinations of PFTs, this model is able to predict some key emergent dynamics of ecosystems based on the underlying biological and evolutionary mechanisms (as shown in Figure 13). Data from field experiments (Ainsworth and Long, 2004; Crowther et al., 2016), observatory networks (e.g., Fluxnet, Baldocchi et al., 2001; Friend et al., 2007), and remote sensing (Duncanson et al., 2020), can provide direct information for modeling the underlying ecological processes and validating predicted emergent properties."

**Minor comments:**

*1. The abbreviation of the term CAI on Line 409 should be put in parentheses for the first time on Line 407.*

Crown area index (CAI). Added.

*2. Lines 71-75: it is unclear to me what is "the legacy of land models and the technical requirements of reversibility in model development"? Could you explain or rephrase this sentence?*

We have reworded this sentence to better clarify our point:

"the long history of land models and the requirements of backward compatibility (i.e., reversing the model to its previous functions) mean developers must often build their new functions on top of previous modeling assumptions and coding structure (Fisher and Koven, 2020), adding up to multiple adjustments of previous processes and making the model untraceable."

*3. Lines 225, "H is tree height" should be modified to "Z is tree height".*

Corrected.

*4. In equation(10), k is ground area? Not defined.*

Yes. We added the explanation to k.

*5. Figure 3. How to understand the constant LAI value of KZ?*

The model defines a maximum LAI for every PFT. In the case of the grass-dominated Konza site, the rapidly growing grass PFT reaches the maximum LAI every year except the first year, when initial density is low. We explained it in the revised manuscript:

"According to the definition of maximum crown LAI (lmax) in Eq. 6, the grass LAI (i.e., Konza) can reach to the maximum each year, except the first year due to the low initial density (Fig. 4: a)."

6. *Figure 9. Please add units of LAI.*

Added.

---

## Author Response (AR2)

Dear Dr. Kato,

We have thoroughly revised our manuscript according to the suggestions of referee 2.

Our responses to each comment are detailed below, with the reviewer comments in italics and our responses in plain text. We thank the reviewers for their thoughtful comments, and the editor for considering our manuscript.

Best regards,

Ensheng Weng (on the behalf of all coauthors)

**Response to referee 1**

No response needed.

**Response to referee 2**

*The authors have made significant improvements based on the reviewers' comments. I enjoyed reading the new manuscript version and am satisfied with their revisions. My major concerns about the previous version are solved. Below are some additional minor suggestions for the authors to further improve their manuscript:*

*Line 162: please add the reference for Beer's law.*

Added:

Beer: Bestimmung der Absorption des rothen Lichts in farbigen Flüssigkeiten, Annalen der Physik, 162, 78–88, https://doi.org/10.1002/andp.18521620505, 1852.

Swinehart, D. F.: The Beer-Lambert Law, J. Chem. Educ., 39, 333, https://doi.org/10.1021/ed039p333, 1962.

*P10, Line 208: although the authors briefly discussed cropland in the discussion, it would be better to mention or explain why they did not include cropland as a plant functional type in the BiomeE model.*

We added one sentence to briefly explain why we didn't include the crop PFTs.

"Cropland is not included because the purpose of this paper is to describe the baseline processes of natural vegetation and soil biogeochemical cycle."

*P16, Line 312: add a full stop.*

Added.

*Figure 5b: It is difficult to differentiate the NBC and TPJ sites.*

We changed the schemes of line types and colors. Accordingly, we updated the figures 3 and 4 with the same color and line type scheme.

*Section 4.2: There are several global data sets of LAI. A quick check on the modeled global pattern of LAI would be helpful in explaining the GPP results.*

Following this suggestion, we added LAI figure for Figures 7 and 8 with the maximum LAI data from Ent vegetation dataset, where the LAI is from a Modis LAI product for the year 2004. We added LAI data sources in the method section, and discussed LAI discrepancies in discussion (5.2 Model predictions and performance).

Copied below is the description of the LAI comparison with data in the result section (4.2 Global Comparisons with Observations):

"The simulated LAI roughly capture the spatial pattern of MODIS LAI (Figure 7: a and b), though there are high variations at each grid (Figure 8: a). Generally, the simulated LAI in well vegetated grids, e.g., boreal forest regions, is underestimated by our model because the crown LAI is calculated as a function of tree height and a parameter of maximum crown LAI (Table 1 and Eq. 6). The LAI in the grids that were converted to different land use types is overestimated because we assume all terrestrial grids are covered by potential vegetation in our test runs."

*Figure 9: A curve of multi-model mean values across the MsTMIP model might be helpful for the comparison.*

Figure 9 is intended to show the spread of MsTMIP model simulations and where BiomeE simulations are. Adding one more dot would complicate the figure.

We suppose the reviewer may refer to the Figure 10. After adding "*A curve of multi-model mean values*", it really looks better. See copied figure 10 below:

[Figure]

*P39: Line 688: LMA has been defined.*

Removed.

*P41: Line 751: add a space after "competition".*

Added.

*P41-42: Some discussions in this paragraph lack references. For example, the "carbon negative leaves" is unclear to most readers. The "uniform leaf" needs more explanation.*

Since we added LAI in the results section, we reorganized the discussion about LAI and crown leaf distribution, added related references, and changed the expressions of "uniform leaves" and "carbon negative leaves" to make them easier to understand (copied below).

"LAI is an illustrative variable for understanding why compromises are necessary when integrating ecological and demographic processes into an ESM. LAI, as a critical prognostic variable in vegetation models, links both plant physiology and biogeophysical interactions with climate systems (Richardson et al., 2012; Kelley et al., 2020; Park and Jeong, 2021). While LAI is usually simulated by a fixed allocation scheme, even if the allocation ratios are dynamic with vegetation productivity or environmental conditions (Montané et al., 2017; Xia et al., 2019), the prediction of LAI is often simplified as the balance between leaf growth and turnover.
In practice, for ESMs, modelers tend to tune the LAI to fit observations and get the required albedo and water fluxes whatever the parameters of photosynthesis and respirations are.  The uniform leaves within a crown would make the lower layer leaves have a negative carbon gain if the LAI was tuned close to that observed in tropical and boreal evergreen forests (around 5~7). Therefore, the photosynthesis rate must be tuned to fit the canopy photosynthesis by keeping these carbon negative leaves.  The crown with carbon negative leaves do not affect the ecosystem carbon dynamics in the "single-cohort" models because the whole canopy net carbon gain can be tuned to fit the observations.
However, for the demographic models, the trees with different sizes are explicitly represented and placed in different layers. The vegetation community can create an understory condition where seedlings cannot survive because of light limitation and negative carbon gains (Weng et al., 2015). Since the leaf traits in the crown profile are functions of light, water and nitrogen (Niinemets et al., 2015), a more complex crown development module is required to simulate branching and leaf development and deployment processes. A tree should be able to optimize its LAI to maximize its fitness as a result of interactions among crown structure, light interception, and community-level competition (Anten, 2002; Niinemets and Anten, 2009; Hikosaka and Anten, 2012).  For balancing the model complexity and computing efficiency, we defined a much small target LAI in this model to avoid carbon negative leaves."

*P46: Line 854: The MsTMIP models were analyzed by Cui et al. 2019 (GBC; 33, 668-689).*

We appreciate the reviewer pointed out the missing reference of MsTMIP and added it in this section. This paper is supportive to what we discussed in this section.

**Additional improvements:**

We edited the whole text thoroughly.

We added a new parameter for PFTs in the table 1, $L_{\max,0}$, since we need this parameter to understand simulated patterns of LAI, which are included in this revised version.

We also updated the Figures S10 and S11 to add the simulated LAI by the single-cohort version of BiomeE.

We improved the discussion of phenology modeling section

"The phenological type is simulated as an emergent property of plant physiological processes and strategies of dealing with seasonal air temperature and soil water variations. Three parameters – growing degree days, running mean daily temperature, and critical soil moisture – are used to define all possible phenological types. These three parameters are widely used in a variety of phenology models (e.g., Prentice et al., 1992; Sitch et al., 2003; Arora and Boer, 2005). However, phenology is not just a physiological response to the seasonality of climate conditions. Evergreen plants are distributed in periodically cold or dry climates. It is a competitively optimal strategy in infertile soil conditions (Aerts, 1995; Givnish, 2002; Coomes et al., 2005). The benefits and costs of keeping different leaves in cold or dry periods should be realistically simulated based on eco-evolutionary theories for phenology modeling (e.g., Levine et al., 2022; Weng et al., 2017)."